# Learning Distributions Generated by One-Layer ReLU Networks

**Shanshan Wu,  Alexandros G. Dimakis,  Sujay Sanghavi**
Department of Electrical and Computer Engineering
University of Texas at Austin
shanshan@utexas.edu, dimakis@austin.utexas.edu, sanghavi@mail.utexas.edu

## Abstract

We consider the problem of estimating the parameters of a $d$-dimensional rectified Gaussian distribution from i.i.d. samples. A rectified Gaussian distribution is defined by passing a standard Gaussian distribution through a one-layer ReLU neural network. We give a simple algorithm to estimate the parameters (i.e., the weight matrix and bias vector of the ReLU neural network) up to an error $\epsilon \|W\|_F$ using $\widetilde{O}(1/\epsilon^2)$ samples and $\widetilde{O}(d^2/\epsilon^2)$ time (log factors are ignored for simplicity). This implies that we can estimate the distribution up to $\epsilon$ in total variation distance using $\widetilde{O}(\kappa^2 d^2/\epsilon^2)$ samples, where $\kappa$ is the condition number of the covariance matrix. Our only assumption is that the bias vector is non-negative. Without this non-negativity assumption, we show that estimating the bias vector within any error requires the number of samples at least exponential in the infinity norm of the bias vector. Our algorithm is based on the key observation that vector norms and pairwise angles can be estimated separately. We use a recent result on learning from truncated samples. We also prove two sample complexity lower bounds: $\Omega(1/\epsilon^2)$ samples are required to estimate the parameters up to error $\epsilon$, while $\Omega(d/\epsilon^2)$ samples are necessary to estimate the distribution up to $\epsilon$ in total variation distance. The first lower bound implies that our algorithm is optimal for parameter estimation. Finally, we show an interesting connection between learning a two-layer generative model and non-negative matrix factorization. Experimental results are provided to support our analysis.

## 1 Introduction

Estimating a high-dimensional distribution from observed samples is a fundamental problem in machine learning and statistics. A popular recent generative approach is to model complex distributions by passing a simple distribution (typically a standard Gaussian) through a neural network. Parameters of the neural network are then learned from data. Generative Adversarial Networks (GANs) [GPAM+14] and Variational Auto-Encoders (VAEs) [KW13] are built on this method of modeling high-dimensional distributions.

Current methods for learning such deep generative models do not have provable guarantees or sample complexity bounds. In this paper we obtain the first such results for a single-layer ReLU generative model. Specifically, we study the following problem: Assume that the latent variable $z$ is selected from a standard Gaussian which then drives the generation of samples from a one-layer ReLU activated neural network with weights $W$ and bias $b$. We observe the output samples (but *not* the latent variable realizations $z$) and we would like to provably learn the parameters $W$ and $b$. More formally:

**Definition 1.** *Let $W \in \mathbb{R}^{d \times k}$ be the weight matrix, and $b \in \mathbb{R}^d$ be the bias vector. We define $\mathcal{D}(W, b)$ as the distribution[1] of the random variable $x \in \mathbb{R}^d$ generated as follows:*

$$x = \mathrm{ReLU}(Wz + b), \text{ where } z \sim \mathcal{N}(0, I_k). \tag{1}$$

*Here $z$ is a standard Gaussian random variable in $\mathbb{R}^k$, and $I_k$ is a $k$-by-$k$ identity matrix.*

Given $n$ samples $x_1, x_2, ..., x_n$ from some $\mathcal{D}(W, b)$ with unknown parameters $W$ and $b$, the goal is to estimate $W$ and $b$ from the given samples. Since the ReLU operation is not invertible[2], estimating $W$ and $b$ via maximum likelihood is often intractable.

In this paper, we make the following contributions:

- We provide a simple and novel algorithm to estimate the parameters of $\mathcal{D}(W, b)$ from i.i.d. samples, under the assumption that $b$ is non-negative. Our algorithm (Algorithm 1) takes two steps. In Step 1, we estimate $b$ and the row norms of $W$ using a recent result on estimation from truncated samples (Algorithm 2). In Step 2, we estimate the angles between any two row vectors of $W$ using a simple geometric result (Fact 1).

- We prove that the proposed algorithm needs $\widetilde{O}(1/\epsilon^2)$ samples and $\widetilde{O}(d^2/\epsilon^2)$ time, in order to estimate the parameter $WW^T$ (reps. $b$) within an error $\epsilon\|W\|_F^2$ (resp. $\epsilon\|W\|_F$) (Theorem 1). This implies that (for the non-degenerate case) the total variation distance between the learned distribution and the ground truth is within an error $\epsilon$ given $\widetilde{O}(\kappa^2 d^2/\epsilon^2)$ samples, where $\kappa$ is the condition number of $WW^T$ (Corollary 1).

- Without the non-negativity assumption on $b$, we show that estimating the parameters of $\mathcal{D}(W, b)$ within any error requires $\Omega(\exp(\|b\|_\infty^2))$ samples (Claim 2). Even when the bias vector $b$ has negative components, our algorithm can still be used to recover part of the parameters with a small amount of samples (Section H.1).

- We prove two lower bounds on the sample complexity. The first lower bound (Theorem 2) says that $\Omega(1/\epsilon^2)$ samples are required in order to estimate $b$ up to error $\epsilon\|W\|_F$, which implies that our algorithm is optimal in estimating the parameters. The second lower bound (Theorem 3) says that $\Omega(d/\epsilon^2)$ samples are required to estimate the distribution up to total variation distance $\epsilon$.

- We empirically evaluate our algorithm in terms of its dependence over the number of samples, dimension, and condition number (Figure 1). The empirical results are consistent with our analysis.

- We provide a new algorithm to estimate the parameters of a two-layer generative model (Algorithm 4 in Appendix H). Our algorithm uses ideas from non-negative matrix factorization (Claim 3).

**Notation.** We use capital letters to denote matrices and lower-case letters to denote vectors. We use $[n]$ to denote the set $\{1, 2, \cdots, n\}$. For a vector $x \in \mathbb{R}^d$, we use $x(i)$ to denote its $i$-th coordinate. The $\ell_p$ norm of a vector is defined as $\|x\|_p = (\sum_i |x(i)|^p)^{1/p}$. For a matrix $W \in \mathbb{R}^{d \times k}$, we use $W(i, j)$ to denote its $(i, j)$-th entry. We use $W(i, :) \in \mathbb{R}^k$ and $W(:, j) \in \mathbb{R}^d$ to the denote the $i$-th row and the $j$-th column. The dot product between two vectors is $\langle x, y \rangle = \sum_i x(i)y(i)$. For any $a \in \mathbb{R}$, we use $\mathbb{R}_{>a}$ to denote the set $\mathbb{R}_{>a} := \{x \in \mathbb{R} : x > a\}$. We use $I_k \in \mathbb{R}^{k \times k}$ to denote an identity matrix.

## 2 Related Work

We briefly review the relevant work, and highlight the differences compared to our paper.

**Estimation from truncated samples.** Given a $d$-dimensional distribution $\mathcal{D}$ and a subset $S \subseteq \mathbb{R}^d$, truncation means that we can only observe samples from $\mathcal{D}$ if it falls in $S$. Samples falling outside $S$ (and their counts in proportion) are not revealed. Estimating the parameters of a multivariate

normal distribution from truncated samples is a fundamental problem in statistics and a breakthrough was achieved recently [DGTZ18] on this problem. This is different from our problem because our samples are formed by *projecting* the samples of a multivariate normal distribution onto the positive orthant instead of *truncating* to the positive orthant. Nevertheless, a single coordinate of $\mathcal{D}(W, b)$ can be viewed as a truncated univariate normal distribution (Definition 2). We use this observation and leverage on the recent results of [DGTZ18] to estimate $b$ and the row norms of $W$ (Section 4.2).

**Learning ReLU neural networks.** A recent series of work, e.g., [GMOV19, GKLW19, GKM18, LY17, ZSJ+17, Sol17], considers the problem of estimating the parameters of a ReLU neural network given samples of the form $\{(x_i, y_i)\}_{i=1}^n$. Here $(x_i, y_i)$ represents the input features and the output target, e.g., $y_i = \text{ReLU}(W x_i + b)$. This is a *supervised* learning problem, and hence, is different from our *unsupervised* density estimation problem.

**Learning neural network-based generative models.** Many approaches have been proposed to train a neural network to model complex distributions. Examples include GAN [GPAM+14] and its variants (e.g., WGAN [ACB17], DCGAN [RMC15], etc.), VAE [KW13], autoregressive models [OKK16], and reversible generative models [GCB+18]. All of those methods lack theoretical guarantees and explicit sample complexity bounds. A recent work [NWH18] proves that training an autoencoder via gradient descent can possibly recover a *linear* generative model. This is different from our setting, where we focus on *non-linear* generative models. Arya and Ankit [MR19] also consider the problem of learning from one-layer ReLU generative models. Their modeling assumption is different from ours. They assume that the bias vector $b$ is a random variable whose distribution satisfies certain conditions. Besides, there is no distributional assumption on the hidden variable $z$. By contrast, in our model, both $W$ and $b$ are deterministic and unknown parameters. The randomness only comes from $z$ which is assumed to follow a standard Gaussian distribution.

## 3 Identifiability

Our first question is whether $W$ is identifiable from the distribution $\mathcal{D}(W, b)$. Claim 1 below implies that only $WW^T$ can be possibly identified from $\mathcal{D}(W, b)$.

**Claim 1.** *For any matrices satisfying $W_1 W_1^T = W_2 W_2^T$, and any vector $b$, $\mathcal{D}(W_1, b) = \mathcal{D}(W_2, b)$.*

*Proof.* Since $W_1 W_1^T = W_2 W_2^T$, there exists a unitary matrix $Q \in \mathbb{R}^{k \times k}$ that satisfies $W_2 = W_1 Q$. Since $z \sim \mathcal{N}(0, I_k)$, we have $Qz \sim \mathcal{N}(0, I_k)$. The claim then follows. $\qquad\square$

Identifying the bias vector $b$ from $\mathcal{D}(W, b)$ can be impossible in some cases. For example, if $W$ is a zero matrix, then any negative coordinate of $b$ cannot be identified since it will be reset to zero after the ReLU operation. For the cases when $b$ is identifiable, our next claim provides a lower bound on the sample complexity required to estimate the bias vector to be within an additive error $\epsilon$.

**Claim 2.** *For any value $\delta > 0$, there exists one-dimensional distributions $\mathcal{D}(1, b_1)$ and $\mathcal{D}(1, b_2)$ such that: (a) $|b_1 - b_2| = \delta$; (b) at least $\Omega(\exp(b_1^2/2))$ samples are required to distinguish them.*

*Proof.* Let $b_1 < 0$ and $b_2 = b_1 - \delta$. It is easy to check that (a) holds. To show (b), note that the probability of observing a positive (i.e., nonzero) sample from $\mathcal{D}(1, b_1)$ is upper bounded by $\mathbb{P}[\text{ReLU}(z - |b_1|) > 0] = \mathbb{P}[z > |b_1|] \leq \exp(-b_1^2/2)$, where the last step follows from the standard Gaussian tail bound [Wai19]. The same bound holds for $\mathcal{D}(1, b_2)$. To distinguish $\mathcal{D}(1, b_1)$ and $\mathcal{D}(1, b_2)$, we need to observe at least one nonzero sample, which requires $\Omega(\exp(b_1^2/2))$ samples. $\quad\square$

Claim 2 indicates that in order to estimate the parameters within any error, the sample complexity should scale at least exponentially in $\|b\|_\infty^2$. This is true if $b$ is allowed to take negative values. Intuitively, if $b$ has large negative values, then most of the samples would be zeros. To avoid this exponential dependence, we now assume that the bias vector is *non-negative*. In Section 4, we give an algorithm to provably learn the parameters of $\mathcal{D}(W, b)$ with a sample complexity that is polynomial in $1/\epsilon$ and does not depend on the values of $b$. In Section H.1, we show that even when the bias vector has negative coordinates, our algorithm can still be able to recover part of the parameters with a small number of samples.

# 4 Algorithm

In this section, we describe a novel algorithm to estimate $WW^T \in \mathbb{R}^{d \times d}$ and $b \in \mathbb{R}^d$ from i.i.d. samples of $\mathcal{D}(W, b)$. Our goal is to estimate $WW^T$ instead of $W$ since $W$ is not identifiable (Claim 1). Our only assumption is that the true $b$ is non-negative. As discussed in Claim 2, this assumption can potentially avoid the exponential dependence in the values of $b$. Note that our algorithm does not require to know the dimension $k$ of the latent variable $z$. Omitted proofs can be found in the appendix.

## 4.1 Intuition

Let $W(i, :) \in \mathbb{R}^k$ be the $i$-th row ($i \in [d]$) of $W$. For any $i < j \in [d]$, the $(i, j)$-th entry of $WW^T$ is

$$\langle W(i, :), W(j, :) \rangle = \|W(i, :)\|_2 \|W(j, :)\|_2 \cos(\theta_{ij}), \tag{2}$$

where $\theta_{ij}$ is the angle between vectors $W(i, :)$ and $W(j, :)$. Our key idea is to estimate the norms $\|W(i, :)\|_2$, $\|W(j, :)\|_2$, and the angles $\theta_{ij}$ separately, as shown in Algorithm 1.

Estimating the row norms[3] $\|W(i, :)\|_2$ as well as the $i$-th coordinate of the bias vector $b(i) \in \mathbb{R}$ can be done by only looking at the $i$-th coordinate of the given samples. The idea is to view the problem as estimating the parameters of a univariate normal distribution from truncated samples[4]. This part of the algorithm is described in Section 4.2. To estimate $\theta_{ij} \in [0, \pi)$ for every $i < j \in [d]$, we use a simple fact that the angle between any two vectors can be estimated from their inner products with a random Gaussian vector. Details of this part can be found in Section 4.3.

---

**Algorithm 1:** Learning a single-layer ReLU generative model

**Input:** $n$ i.i.d. samples $x_1, \cdots, x_n \in \mathbb{R}^d$ from $\mathcal{D}(W^*, b^*)$, $b^*$ is non-negative.
**Output:** $\widehat{\Sigma} \in \mathbb{R}^{d \times d}$, $\widehat{b} \in \mathbb{R}^d$.

1 **for** $i \leftarrow 1$ **to** $d$ **do**
2      $S \leftarrow \{x_m(i), m \in [n] : x_m(i) > 0\}$;
3      $\widehat{b}(i), \widehat{\Sigma}(i, i) \leftarrow \text{NormBiasEst}(S)$;
4      $\widehat{b}(i) \leftarrow \max\left(0, \widehat{b}(i)\right)$;
5 **end**
6 **for** $i < j \in [d]$ **do**
7      $\widehat{\theta}_{ij} \leftarrow \pi - \frac{2\pi}{n}\left(\sum_{m=1}^{n} \mathbb{1}(x_m(i) > \widehat{b}(i))\, \mathbb{1}(x_m(j) > \widehat{b}(j))\right)$;
8      $\widehat{\Sigma}(i, j) \leftarrow \sqrt{\widehat{\Sigma}(i, i)\widehat{\Sigma}(j, j)} \cos(\widehat{\theta}_{ij})$;
9      $\widehat{\Sigma}(j, i) \leftarrow \widehat{\Sigma}(i, j)$;
10 **end**

---

## 4.2 Estimate $\|W(i, :)\|_2$ and $b(i)$

Without loss of generality, we fix $i = 1$ and describe how to estimate $\|W(1, :)\|_2 \in \mathbb{R}$ and $b(1) \in \mathbb{R}$ by looking at the first coordinate of the given samples.

The starting point of our algorithm is the following observation. Suppose $x \sim \mathcal{D}(W, b)$, its first coordinate can be written as

$$x(1) = \text{ReLU}(W(1, :)^T z + b(1)) = \text{ReLU}(y), \text{ where } y \sim \mathcal{N}(b(1), \|W(1, :)\|_2^2). \tag{3}$$

Because of the ReLU operation, we can only observe the samples of $y$ when it is positive. Given samples of $x(1) \in \mathbb{R}$, let us keep the samples that have positive values (i.e., ignore the zero samples).

Now the problem of estimating $b(1)$ and $\|W(1,:)\|_2$ is equivalent to estimating the parameters of a one-dimensional normal distribution using samples falling in the set $\mathbb{R}_{>0} := \{x \in \mathbb{R} : x > 0\}$.

Recently Daskalakis et al. [DGTZ18] gave an efficient algorithm for estimating the mean and covariance matrix of a multivariate Gaussian distribution from truncated samples. We adapt their algorithm for the specific problem described above. Before describing the details, we start with a formal definition of the truncated (univariate) normal distribution.

**Definition 2.** *The univariate normal distribution $\mathcal{N}(\mu, \sigma^2)$ has probability density function*

$$\mathcal{N}(\mu, \sigma^2; x) = \frac{1}{\sqrt{2\pi\sigma^2}} \exp\left(-\frac{1}{2\sigma^2}(x-\mu)^2\right), \quad \text{for } x \in \mathbb{R}. \tag{4}$$

*Given a measurable set $S \subseteq \mathbb{R}$, the $S$-truncated normal distribution $\mathcal{N}(\mu, \sigma^2, S)$ is defined as*

$$\mathcal{N}(\mu, \sigma^2, S; x) = \begin{cases} \frac{\mathcal{N}(\mu,\sigma^2;x)}{\int_S \mathcal{N}(\mu,\sigma^2;y)dy} & \text{if } x \in S \\ 0 & \text{if } x \notin S \end{cases}. \tag{5}$$

We are now ready to describe the algorithm in [DGTZ18] applied to our problem. The pseudocode is given in Algorithm 2. The algorithm is essentially maximum likelihood by projected stochastic gradient descent (SGD). Given a sample $x \sim \mathcal{N}(\mu^*, \sigma^{*2}, S)$, let $\ell(\mu, \sigma; x)$ be the negative log-likelihood that $x$ is from $\mathcal{N}(\mu, \sigma^2, S)$, then $\ell(\mu, \sigma; x)$ is a convex function with respect to a reparameterization $v = [1/\sigma^2, \mu/\sigma^2] \in \mathbb{R}^2$. We use $\ell(v; x)$ to denote the negative log-likelihood after this reparameterization. Let $\bar{\ell}(v) = \mathbb{E}_x[\ell(v; x)]$ be the expected negative log-likelihood. Although it is intractable to compute $\bar{\ell}(v)$, its gradient $\nabla \bar{\ell}(v)$ with respect to $v$ has a simple unbiased estimator. Specifically, define a random vector $g \in \mathbb{R}^2$ as

$$g = -\begin{bmatrix} -x^2/2 \\ x \end{bmatrix} + \begin{bmatrix} -z^2/2 \\ z \end{bmatrix}, \text{ where } x \sim \mathcal{N}(\mu^*, \sigma^{*2}, S), z \sim \mathcal{N}(\mu, \sigma^2, S). \tag{6}$$

We have that $\nabla \bar{\ell}(v) = \mathbb{E}_{x,z}[g]$, i.e., $g$ is an unbiased estimator of $\nabla \bar{\ell}(v)$.

Eq. (6) indicates that one can maximize the log-likelihood via SGD, however, in order to efficiently perform this optimization, we need three extra steps.

First, the convergence rate of SGD depends on the expected gradient norm $\mathbb{E}[\|g\|_2^2]$ (Theorem 14.11 of [SSBD14]). In order to maintain a small gradient norm, we transform the given samples to a new space (so that the empirical mean and variance is well-controlled) and perform optimization in that space. After the optimization is done, the solution is transformed back to the original space. Specifically, given samples $x_1, \cdots, x_n \sim \mathcal{N}(\mu^*, \sigma^{*2}, \mathbb{R}_{>0})$, we transform them as

$$x_i \rightarrow \frac{x_i - \widehat{\mu}_0}{\widehat{\sigma}_0}, \text{ where } \widehat{\mu}_0 = \frac{1}{n} \sum_{i=1}^n x_i, \ \widehat{\sigma}_0^2 = \frac{1}{n} \sum_{i=1}^n (x_i - \widehat{\mu}_0)^2. \tag{7}$$

In the transformed space, the problem becomes estimating parameters of a normal distribution with samples truncated to the set $\mathbb{R}_{>-\widehat{\mu}_0/\widehat{\sigma}_0} = \{x \in \mathbb{R} : x > -\widehat{\mu}_0/\widehat{\sigma}_0\}$.

Second, we need to control the strong-convexity of the objective function. This is done by projecting the parameters onto a domain where the strong-convexity is bounded. The domain $D_r$ is parameterized by $r > 0$ and is defined as

$$D_r = \{v \in \mathbb{R}^2 : 1/r \leq v(1) \leq r, |v(2)| \leq r\}. \tag{8}$$

According to [DGTZ18, Section 3.4], $r = O(\ln(1/\alpha)/\alpha^2)$ is a hyper-parameter that only depends on $\alpha = \int_S \mathcal{N}(\mu^*, \sigma^{*2}; y)dy$ (i.e., the probability mass of original truncation set $S$). In our setting, we have $\alpha \geq 1/2$. This is because the original truncation set is $\mathbb{R}_{>0}$ and $\mu^* = b(1) \geq 0$. A large value of $r$ would lead to a small strong-convexity parameter. In our experiments, we set $r = 3$.

Third, a single run of the projected SGD algorithm only guarantees a constant probability of success. To amplify the probability of success to $1 - \delta/d$, a standard procedure is to repeat the algorithm $O(\ln(d/\delta))$ times. This procedure is illustrated in Step 2-5 in Algorithm 2.

**Algorithm 2:** NormBiasEst

**Input:** Samples from $\mathcal{N}(\mu, \sigma^2, \mathbb{R}_{>0})$.

**Output:** $\widehat{\mu} \in \mathbb{R}, \widehat{\sigma^2} \in \mathbb{R}$.

**1** Shift and rescale the samples using (7);

**2** Split the samples into $B = O(\ln(d/\delta))$ batches;

**3** For batch $i \in [B]$, run ProjSGD to get $v_i \in \mathbb{R}^2$;

**4** $S \leftarrow \{v_1, \cdots, v_B\}$;

**5** $\widehat{v} \leftarrow \arg\min_{v_i \in S} \sum_{j \in [B]} \|v_i - v_j\|_2$;

**6** Transform $\widehat{v}$ back to the original space;

**7** $\widehat{\mu} \leftarrow \widehat{v}(2)/\widehat{v}(1), \widehat{\sigma^2} \leftarrow 1/\widehat{v}(1)$;

**Algorithm 3:** ProjSGD

**Input:** $T = \widetilde{O}(\ln(d/\delta)/\epsilon^2), \lambda > 0$.

**Output:** $v \in \mathbb{R}^2$.

**1** Initialize $v^{(0)} = [1, 0] \in \mathbb{R}^2$;

**2** for $t \leftarrow 1$ to $T$ do

**3** $\quad g^{(t)} \leftarrow$ Estimate the gradient using (6);

**4** $\quad v^{(t)} \leftarrow v^{(t-1)} - g^{(t)}/(\lambda \cdot t)$;

**5** $\quad v^{(t)} \leftarrow$ Project $v^{(t)}$ to the domain in (8);

**6** end

**7** $v \leftarrow \sum_{t=1}^{T} v^{(t)}/T$;

**Lemma 1.** *For any $\epsilon \in (0,1)$ and $\delta \in (0,1)$, Algorithm 1 takes $n = \widetilde{O}\left(\frac{1}{\epsilon^2}\ln(\frac{d}{\delta})\right)$ samples from $\mathcal{D}(W^*, b^*)$ (for some non-negative $b^*$) and outputs $\widehat{b}(i)$ and $\widehat{\Sigma}(i,i)$ for all $i \in [d]$ that satisfy*

$$(1-\epsilon)\|W^*(i,:)\|_2^2 \leq \widehat{\Sigma}(i,i) \leq (1+\epsilon)\|W^*(i,:)\|_2^2, \quad |\widehat{b}(i) - b^*(i)| \leq \epsilon\|W^*(i,:)\|_2 \qquad (9)$$

*with probability at least $1 - \delta$.*

### 4.3 Estimate $\theta_{ij}$

To estimate the angle between any two vectors $W^*(i,:)$ and $W^*(j,:)$ (where $i \neq j \in [d]$), we will use the following result.

**Fact 1.** *(Lemma 6.7 in [WS11]). Let $z \sim \mathcal{N}(0, I_k)$ be a standard Gaussian random variable in $\mathbb{R}^k$. For any two non-zero vectors $u, v \in \mathbb{R}^k$, the following holds:*

$$\mathbb{P}_{z \sim \mathcal{N}(0, I_k)}[u^T z > 0 \text{ and } v^T z > 0] = \frac{\pi - \theta}{2\pi}, \text{ where } \theta = \arccos\left(\frac{\langle u, v \rangle}{\|u\|_2\|v\|_2}\right). \qquad (10)$$

Fact 1 says that the angle between any two vectors can be estimated from the sign of their inner products with a Gaussian random vector. Let $x \sim \mathcal{D}(W^*, b^*)$, since $b^*$ is assumed to be non-negative, Fact 1 gives an unbiased estimator for the pairwise angles.

**Lemma 2.** *Suppose that $x \sim \mathcal{D}(W^*, b^*)$ and that $b^* \in \mathbb{R}^d$ is non-negative, for all $i \neq j \in [d]$,*

$$\mathbb{P}_{x \sim \mathcal{D}(W^*, b^*)}[x(i) > b^*(i) \text{ and } x(j) > b^*(j)] = \frac{\pi - \theta_{ij}^*}{2\pi}, \qquad (11)$$

*where $\theta_{ij}^*$ is the angle between vectors $W^*(i,:)$ and $W^*(j,:)$.*

*Proof.* Since $x(i) = \text{ReLU}\left(W^*(i,:)^T z + b^*(i)\right)$ and $b^*$ is non-negative, we have

$$\text{LHS} = \mathbb{P}_{z \sim \mathcal{N}(0, I_k)}[W^*(i,:)^T z > 0 \text{ and } W^*(j,:)^T z > 0] = \frac{\pi - \theta_{ij}^*}{2\pi} = \text{RHS}, \qquad (12)$$

where the second equality follows from Fact 1. $\qquad \square$

Lemma 2 gives an unbiased estimator of $\theta_{ij}^*$, however, it requires knowing the true bias vector $b^*$. In the previous section, we give an algorithm that can estimate $b^*(i)$ within an additive error of $\epsilon\|W^*(i,:)\|_2$ for all $i \in [d]$. Fortunately, this is good enough for estimating $\theta_{ij}^*$ within an additive error of $\epsilon$, as indicated by the following lemma.

**Lemma 3.** *Let $x \sim \mathcal{D}(W^*, b^*)$, where $b^*$ is non-negative. Suppose that $\widehat{b} \in \mathbb{R}^d$ is non-negative and satisfies $|\widehat{b}(i) - b^*(i)| \leq \epsilon\|W^*(i,:)\|_2$ for all $i \in [d]$ and some $\epsilon > 0$. Then for all $i \neq j \in [d]$,*

$$\left|\mathbb{P}_x[x(i) > \widehat{b}(i) \text{ and } x(j) > \widehat{b}(j)] - \mathbb{P}_x[x(i) > b^*(i) \text{ and } x(j) > b^*(j)]\right| \leq \epsilon. \qquad (13)$$

Let $\mathbb{1}(\cdot)$ be the indicator function, e.g., $\mathbb{1}(x > 0) = 1$ if $x > 0$ and is 0 otherwise. Given samples $\{x_m\}_{m=1}^n$ of $\mathcal{D}(W^*, b^*)$ and an estimated bias vector $\widehat{b}$, Lemma 2 and 3 implies that $\theta_{ij}^*$ can be estimated as

$$\widehat{\theta}_{ij} = \pi - \frac{2\pi}{n} \sum_{m=1}^n \mathbb{1}(x_m(i) > \widehat{b}(i) \text{ and } x_m(j) > \widehat{b}(j)). \tag{14}$$

The following lemma shows that the estimated $\widehat{\theta}_{ij}$ is close to the true $\theta_{ij}^*$.

**Lemma 4.** *For a fixed pair of $i \neq j \in [d]$, for any $\epsilon, \delta \in (0, 1)$, suppose $\widehat{b}$ satisfies the condition in Lemma 3, given $80 \ln(2/\delta)/\epsilon^2$ samples, with probability at least $1 - \delta$, $|\cos(\widehat{\theta}_{ij}) - \cos(\theta_{ij}^*)| \leq \epsilon$.*

## 4.4 Estimate $WW^T$ and $b$

Our overall algorithm is given in Algorithm 1. In the first for-loop, we estimate the row norms of $W^*$ and $b^*$. In the second for-loop, we estimate the angles between any two row vectors of $W^*$.

**Theorem 1.** *For any $\epsilon \in (0, 1)$ and $\delta \in (0, 1)$, Algorithm 1 takes $n = \widetilde{O}\left(\frac{1}{\epsilon^2} \ln(\frac{d}{\delta})\right)$ samples from $\mathcal{D}(W^*, b^*)$ (for some non-negative $b^*$) and outputs $\widehat{\Sigma} \in \mathbb{R}^{d \times d}$ and $\widehat{b} \in \mathbb{R}^d$ that satisfy*

$$\|\widehat{\Sigma} - W^* W^{*T}\|_F \leq \epsilon \|W^*\|_F^2, \quad \|\widehat{b} - b^*\|_2 \leq \epsilon \|W^*\|_F \tag{15}$$

*with probability at least $1 - \delta$. Algorithm 1 runs in time $\widetilde{O}\left(\frac{d^2}{\epsilon^2} \ln(\frac{d}{\delta})\right)$ and space $\widetilde{O}\left(\frac{d}{\epsilon^2} \ln(\frac{d}{\delta}) + d^2\right)$.*

Theorem 1 characterizes the sample complexity to achieve a small parameter estimation error. We are also interested in the distance between the estimated distribution and the true distribution. Let $\text{TV}(A, B)$ be the total variation (TV) distance between two distributions $A$ and $B$. Note that in order for the TV distance to be meaningful[5], we restrict ourselves to the non-degenerate case, i.e., when $W$ is a full-rank square matrix. The following corollary characterizes the number of samples used by our algorithm in order to achieve a small TV distance.

**Corollary 1.** *Suppose that $W^* \in \mathbb{R}^{d \times d}$ is full-rank. Let $\kappa$ be the condition number of $W^* W^{*T}$. For any $\epsilon \in (0, 1/2]$ and $\delta \in (0, 1)$, Algorithm 1 takes $n = \widetilde{O}\left(\frac{\kappa^2 d^2}{\epsilon^2} \ln(\frac{d}{\delta})\right)$ samples from $\mathcal{D}(W^*, b^*)$ (for some non-negative $b^*$) and outputs a distribution $\mathcal{D}(\widehat{\Sigma}^{1/2}, \widehat{b})$ that satisfies*

$$\text{TV}\left(\mathcal{D}(\widehat{\Sigma}^{1/2}, \widehat{b}), \ \mathcal{D}(W^*, b^*)\right) \leq \epsilon, \tag{16}$$

*with probability at least $1 - \delta$. Algorithm 1 runs in time $\widetilde{O}\left(\frac{\kappa^2 d^4}{\epsilon^2} \ln(\frac{d}{\delta})\right)$ and space $\widetilde{O}\left(\frac{\kappa^2 d^3}{\epsilon^2} \ln(\frac{d}{\delta})\right)$.*

# 5 Lower Bounds

In the previous section, we gave an algorithm to estimate $W^* W^{*T}$ and $b^*$ using i.i.d. samples from $\mathcal{D}(W^*, b^*)$, and analyzed its sample complexity. In this section, we provide lower bounds for this density estimation problem. More precisely, we want to know: how many samples are necessary if we want to learn $\mathcal{D}(W^*, b^*)$ up to some error measure $\epsilon$?

Before stating our lower bounds, we first formally define a framework for distribution learning[6]. Let $S$ be a class of distributions. Let $d$ be some distance function between the two distributions (or between the parameters of the two distributions). We say that a distribution learning algorithm learns $S$ with sample complexity $m(\epsilon)$ if for any distribution $p \in S$, given $m(\epsilon)$ i.i.d. samples from $p$, it constructs a distribution $q$ such that $d(p, q) \leq \epsilon$ with success probability at least $2/3$[7].

We have analyzed the performance of Algorithm 1 in terms of two distance metrics: the distance in the parameter space (Theorem 1), and the TV distance between two distributions (Corollary 1). Accordingly, we will provide two sample complexity lower bounds.

**Theorem 2.** *(Lower bound for parameter estimation). Let $\sigma > 0$ be a fixed and known scalar. Let $I_d$ be the identity matrix in $\mathbb{R}^d$. Let $S := \{\mathcal{D}(W, b) : W = \sigma I_d, b \in \mathbb{R}^d$ non-negative$\}$ be a class of distributions in $\mathbb{R}^d$. Any algorithm that learns $S$ to satisfy $\|\widehat{b} - b^*\|_2 \leq \epsilon \|W^*\|_F$ with success probability at least 2/3 requires $\Omega(1/\epsilon^2)$ samples.*

**Theorem 3.** *(Lower bound for distribution estimation). Let $S := \{\mathcal{D}(W, 0) : W \in \mathbb{R}^{d \times d}$ full rank$\}$ be a set of distributions in $\mathbb{R}^d$. Any algorithm that learns $S$ within total variation distance $\epsilon$ and success probability at least 2/3 requires $\Omega(d/\epsilon^2)$ samples.*

Comparing the sample complexity achieved by our algorithm (Theorem 1 and Corollary 1) and the above lower bounds, we can see that 1) our algorithm matches the lower bound (up to log factors) for parameter estimation; 2) there is a gap between our sample complexity and the lower bound for TV distance. There are two possible reasons why this gap shows up.

- The lower bound given in Theorem 3 may be loose. In fact, since learning a $d$-dimensional Gaussian distribution up to TV distance $\epsilon$ requires $\widetilde{\Theta}(d^2/\epsilon^2)$ samples (this is both sufficient and necessary [ABDH+18]), it is reasonable to guess that learning rectified Gaussian distributions also requires at least $\Omega(d^2/\epsilon^2)$ samples. It is thus interesting to see if one can show a better lower bound than $\Omega(d/\epsilon^2)$.

- Our sample complexity of learning $\mathcal{D}(W, b)$ up to TV distance $\epsilon$ also depends on the condition number $\kappa$ of $WW^T$. Intuitively, this $\kappa$ dependence shows up because our algorithm estimates $WW^T$ entry-by-entry instead of estimating the matrix as a whole. Besides, our algorithm is a *proper* learning algorithm, meaning that the output distribution belongs to the family $\mathcal{D}(W, b)$. By contrast, the lower bound proved in Theorem 3 considers any *non-proper* learning algorithm, i.e., there is no constraint on the output distribution. One interesting direction for future research is to see if one can remove this $\kappa$ dependence.

# 6 Experiments

In this section, we provide empirical results to verify the correctness of our algorithm as well as the analysis. Code to reproduce our result[8] can be found at `https://github.com/wushanshan/densityEstimation`.

We evaluate three performance metrics, as shown in Figure 1. The first two metrics measure the error between the estimated parameters and the ground truth. Specifically, we compute the estimation errors analyzed in Theorem 1: $\|\widehat{\Sigma} - W^* W^{*T}\|_F / \|W\|_F^2$ and $\|\widehat{b} - b\|_2 / \|W\|_F$. Besides the parameter estimation error, we are also interested in the TV distance analyzed in Corollary 1: $\mathrm{TV}\left(\mathcal{D}(\widehat{\Sigma}^{1/2}, \widehat{b}), \mathcal{D}(W^*, b^*)\right)$. It is difficult to compute the TV distance exactly, so we instead compute an upper bound of it. Let $KL(A\|B)$ denote the KL divergence between two distributions. Let $\Sigma^* = W^* W^{*T}$. Assuming that both $\Sigma^*$ and $\widehat{\Sigma}$ are full-rank, we have

$$\mathrm{TV}\left(\mathcal{D}(\widehat{\Sigma}^{1/2}, \widehat{b}), \mathcal{D}(W^*, b^*)\right) \leq \mathrm{TV}\left(\mathcal{N}(\widehat{b}, \widehat{\Sigma}), \mathcal{N}(b^*, \Sigma^*)\right) \leq \sqrt{KL\left(\mathcal{N}(\widehat{b}, \widehat{\Sigma})\|\mathcal{N}(b^*, \Sigma^*)\right)/2}.$$

The first inequality follows from the data-processing inequality given in Lemma 7 of Appendix F (see also [ABDH+18, Fact A.5]): for any function $f$ and random variables $X, Y$ over the same space, $\mathrm{TV}(f(X), f(Y)) \leq \mathrm{TV}(X, Y)$. The second inequality follows from the Pinsker's inequality [Tsy09, Lemma 2.5].

**Sample Efficiency.** The left plot of Figure 1 shows that both the parameter estimation errors and the KL divergence decrease when we have more samples. Our experimental setting is simple: we set the dimension as $d = k = 5$ and the condition number as 1; we generate $W^*$ as a random orthonormal matrix; we generate $b^*$ as a random normal vector, followed by a ReLU operation (to ensure non-negativity). This plot indicates that our algorithm is able to accurately estimate the true parameters and obtain a distribution that is close to the true distribution in TV distance.

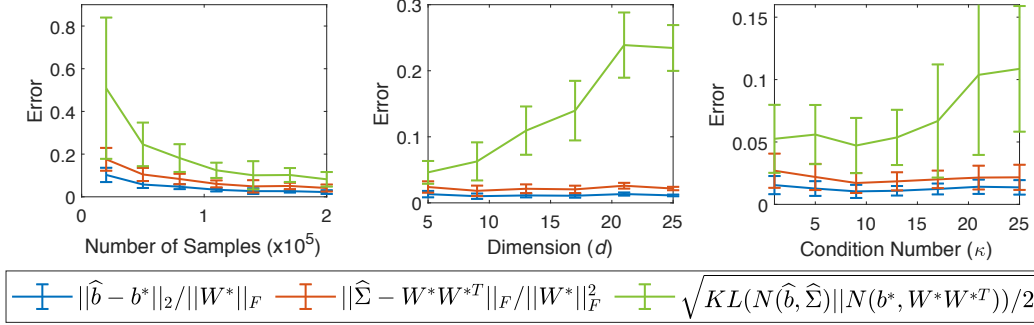

Figure 1: Best viewed in color. Empirical performance of our algorithm with respect to three parameters: number of samples $n$, dimension $d$, and the condition number $\kappa$. **Left:** Fix $d = 5$ and $\kappa = 1$. **Middle:** Fix $n = 5 \times 10^5$ and $\kappa = 1$. **Right:** Fix $n = 5 \times 10^5$ and $d = 5$. Every point shows the mean and standard deviation across 10 runs. Each run corresponds to a different $W^*$ and $b^*$.

**Dependence on Dimension.** In the middle plot of Figure 1, we use $5 \times 10^5$ samples and keep the condition number to be 1. We then increase the dimension ($d = k$) from 5 to 25. Both $W^*$ and $b^*$ are generated in the same manner as the previous plot. As shown in the middle plot, the parameter estimation errors maintain the same value while the KL divergence increases as the dimension increases. This is consistent with our analysis, because the sample complexity in Theorem 1 is dimension-free (ignoring the log factor) while the sample complexity in Corollary 1 depends on $d^2$.

**Dependence on Condition Number.** In the right plot of Figure 1, we keep the dimension $d = k = 5$ and the number of samples $5 \times 10^5$ fixed. We then increase the condition number $\kappa$ of $W^*W^{*T}$. This plot shows the same trend as the middle plot, i.e., the parameter estimation errors remain the same while the KL divergence increases as $\kappa$ increases, which is again consistent with our analysis. The number of samples required to achieve an additive estimation error (Theorem 1) does not depend on $\kappa$, while the sample complexity to guarantee a small TV distance (Corollary 1) depends on $\kappa^2$.

# 7    Conclusion

A popular generative model nowadays is defined by passing a standard Gaussian random variable through a neural network. In this paper we are interested in the following fundamental question: Given samples from this distribution, is it possible to recover the parameters of the underlying neural network? We designed a new algorithm to provably recover the parameters of a single-layer ReLU generative model from i.i.d. samples, under the assumption that the bias vector is non-negative. We analyzed the sample complexity of the proposed algorithm in terms of two error metrics: parameter estimation error and total variation distance. Sample complexity lower bounds and experimental results are provided to support our analysis.

There are many questions that one could ask here. For example, what happens if the bias vector has negative values? What if the generative model has two layers? What if the samples are noisy? We summarized our thoughts on some problems in Appendix H. In particular, we showed an interesting connection between learning a two-layer generative model and non-negative matrix factorization.

While our focus here is parameter recovery, one interesting direction for future work is to see whether one can directly estimate the distribution in some distance without first estimating the parameters. Another interesting direction is to develop provable learning algorithms for the agnostic setting instead of the realizable setting. Besides designing new algorithms, analyzing the existing algorithms, e.g., GANs, VAEs, and reversible generative models, is also an important research direction.

# 8    Acknowledgements

This research has been supported by NSF Grants 1302435, 1564000, and 1618689, DMS 1723052, CCF 1763702, AF 1901292 and research gifts by Google, Western Digital and NVIDIA.

## Footnotes

[1] It is also called as a rectified Gaussian distribution, and can be used in non-negative factor analysis [HK07].

[2] If the activation function $\sigma$ (e.g., sigmoid, leaky ReLU, etc.) is invertible, then $\sigma^{-1}(X) \sim \mathcal{N}(b, WW^T)$. In that case the problem becomes learning a Gaussian from samples.

[3]Without loss of generality, we can assume that $\|W(i, :)\|_2 \neq 0$ for all $i \in [d]$. If $W(i, :)$ is a zero vector, one can easily detect that and figure out the corresponding non-negative bias term.

[4]Another idea is to use the median of the samples to estimate the $i$-th coordinate of the bias vector. This approach will give the same sample complexity bound as that of our proposed algorithm.

[5]The TV distance between two different degenerate distributions can be a constant. As an example, let $\mathcal{N}(0, \Sigma_1)$ and $\mathcal{N}(0, \Sigma_2)$ be two Gaussian distributions in $\mathbb{R}^d$. If both $\Sigma_1, \Sigma_2$ have rank smaller than $d$, then $\text{TV}(\mathcal{N}(0, \Sigma_1), \mathcal{N}(0, \Sigma_2)) = 1$ as long as $\Sigma_1 \neq \Sigma_2$.

[6]This can be viewed as the standard PAC-learning framework [Val84].

[7]We focus on constant success probability here as standard techniques can be used to boost the success probability to $1 - \delta$ with an extra multiplicative factor $\ln(1/\delta)$ in the sample complexity.

[8]The hyper-parameters are $B = 1$ (in Algorithm 2), $r = 3$ and $\lambda = 0.1$ (in Algorithm 3).

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
