[Supplementary Material]

# A  Proof of Lemma 1

We first restate the lemma and then give the proof.

**Lemma.** *For any $\epsilon \in (0,1)$ and $\delta \in (0,1)$, Algorithm 1 takes $n = \widetilde{O}\left(\frac{1}{\epsilon^2}\ln(\frac{d}{\delta})\right)$ samples from $\mathcal{D}(W^*, b^*)$ (for some non-negative $b^*$) and outputs $\widehat{b}(i)$ and $\widehat{\Sigma}(i,i)$ for all $i \in [d]$ that satisfy*

$$(1 - \epsilon)\|W^*(i,:)\|_2^2 \leq \widehat{\Sigma}(i,i) \leq (1 + \epsilon)\|W^*(i,:)\|_2^2, \quad |\widehat{b}(i) - b^*(i)| \leq \epsilon\|W^*(i,:)\|_2$$

*with probability at least $1 - \delta$.*

*Proof.* For a fixed $i \in [d]$, according to Theorem 1 of [DGTZ18], given $\widetilde{O}(\ln(d/\delta)/\epsilon^2)$ truncated samples from $\mathcal{N}(b^*(i), \|W^*(i,:)\|_2^2, \mathbb{R}_{>0})$, the output of Algorithm 2 satisfies (9) with probability at least $1 - \delta/d$. Since $b^*(i) \geq 0$, a sample $x \sim \mathcal{N}(b^*(i), \|W^*(i,:)\|_2^2)$ satisfies $x > 0$ with probability at least $1/2$. By Hoeffding's inequality, if we take $\widetilde{O}(\ln(d/\delta)/\epsilon^2) + O(\ln(d/\delta)) = \widetilde{O}(\ln(d/\delta)/\epsilon^2)$ samples from $\mathcal{D}(W^*, b^*)$, then we are able to obtain $\widetilde{O}(\ln(d/\delta)/\epsilon^2)$ truncated samples with probability at least $1 - \delta/d$. Therefore, if we take $\widetilde{O}(\ln(d/\delta)/\epsilon^2)$ samples from $\mathcal{D}(W^*, b^*)$, for a fixed coordinate $i \in [d]$, the output of Algorithm 1 satisfies (9) with probability at least $1 - 2\delta/d$. Lemma 1 then follows by taking a union bound over all coordinates in $[d]$ and re-scaling $\delta$ to $\delta/2$. $\qquad\square$

# B  Proof of Lemma 3

We first restate the lemma and then give the proof.

**Lemma.** *Let $x \sim \mathcal{D}(W^*, b^*)$, where $b^*$ is non-negative. Suppose that $\widehat{b} \in \mathbb{R}^d$ is non-negative and satisfies $|\widehat{b}(i) - b^*(i)| \leq \epsilon\|W^*(i,:)\|_2$ for all $i \in [d]$ and some $\epsilon > 0$. Then for all $i \neq j \in [d]$,*

$$\left|\mathbb{P}_x[x(i) > \widehat{b}(i) \text{ and } x(j) > \widehat{b}(j)] - \mathbb{P}_x[x(i) > b^*(i) \text{ and } x(j) > b^*(j)]\right| \leq \epsilon.$$

*Proof.* We first notice that $\widehat{b}$ satisfies

$$\max(0, b^*(i) - \epsilon\|W^*(i,:)\|_2) \leq \widehat{b}(i) \leq b^*(i) + \epsilon\|W^*(i,:)\|_2, \text{ for all } i \in [d]. \tag{17}$$

To prove Lemma 3, we only need to prove that (13) holds when $\widehat{b}$ is substituted by its lower bound as well as the upper bound. We focus on substituting the lower bound here (as the upper bound follows a similar proof). We assume that $\|W^*(i,:)\|_2 \neq 0$ for all $i \in [d]$ (the proof extends straightforwardly to the setting when this is not true).

$$\mathbb{P}_x\left[x(i) > \max(0, b^*(i) - \epsilon\|W^*(i,:)\|_2) \text{ and } x(j) > \max(0, b^*(j) - \epsilon\|W^*(j,:)\|_2)\right]$$

$$- \mathbb{P}_x[x(i) > b^*(i) \text{ and } x(j) > b^*(j)]$$

$$\overset{(a)}{\leq} \mathbb{P}_{z \sim \mathcal{N}(0, I_k)}\left[W^*(i,:)^T z > -\epsilon\|W^*(i,:)\|_2 \text{ and } W^*(j,:)^T z > -\epsilon\|W^*(j,:)\|_2\right]$$

$$- \mathbb{P}_{z \sim \mathcal{N}(0, I_k)}[W^*(i,:)^T z > 0 \text{ and } W^*(j,:)^T z > 0]$$

$$= \mathbb{P}_{z \sim \mathcal{N}(0, I_k)}\left[-\epsilon < \frac{W^*(i,:)^T}{\|W^*(i,:)\|_2} z \leq 0 \text{ and } -\epsilon < \frac{W^*(j,:)^T}{\|W^*(j,:)\|_2} z \leq 0\right]$$

$$+ \mathbb{P}_{z \sim \mathcal{N}(0, I_k)}\left[-\epsilon < \frac{W^*(i,:)^T}{\|W^*(i,:)\|_2} z \leq 0 \text{ and } \frac{W^*(j,:)^T}{\|W^*(j,:)\|_2} z > 0\right]$$

$$+ \mathbb{P}_{z \sim \mathcal{N}(0, I_k)}\left[\frac{W^*(i,:)^T}{\|W^*(i,:)\|_2} z > 0 \text{ and } -\epsilon < \frac{W^*(j,:)^T}{\|W^*(j,:)\|_2} z \leq 0\right]$$

$$\leq \mathbb{P}_{z \sim \mathcal{N}(0, I_k)}\left[-\epsilon < \frac{W^*(i,:)^T}{\|W^*(i,:)\|_2} z \leq 0\right] + \mathbb{P}_{z \sim \mathcal{N}(0, I_k)}\left[-\epsilon < \frac{W^*(j,:)^T}{\|W^*(j,:)\|_2} z \leq 0\right]$$

$$\overset{(b)}{\leq} \frac{2}{\sqrt{2\pi}}\epsilon \leq \epsilon.$$

Here (a) is true because $x(i) = \text{ReLU}\left(W^*(i,:)^T z + b^*(i)\right)$ and $b^*$ is non-negative. Inequality (b) is true because $\frac{W^*(i,:)^T}{\|W^*(i,:)\|_2} z$ is a one-dimensional Gaussian distribution $\mathcal{N}(0,1)$ and the probability density of $\mathcal{N}(0,1)$ has value no larger than $1/\sqrt{2\pi}$. $\qquad\square$

## C  Proof of Lemma 4

We first restate the lemma and then give the proof.

**Lemma.** *For a fixed pair of $i \neq j \in [d]$, for any $\epsilon, \delta \in (0,1)$, suppose $\widehat{b}$ satisfies the condition in Lemma 3, given $80 \ln(2/\delta)/\epsilon^2$ samples, with probability at least $1 - \delta$, $|\cos(\widehat{\theta}_{ij}) - \cos(\theta_{ij})| \leq \epsilon$.*

*Proof.* For a fixed pair $i \neq j \in [d]$, let $f(x) := \mathbb{1}(x(i) > \widehat{b}(i) \text{ and } x(j) > \widehat{b}(j))$. Since the indicator function is bounded, Hoeffding's inequality implies that if the number of samples $n \geq \ln(2/\delta)/(2\epsilon^2)$, then with probability at least $1 - \delta$,

$$\left| \frac{1}{n} \sum_{m=1}^n f(x_m) - \mathbb{E}_x[f(x)] \right| \leq \epsilon. \tag{18}$$

By Lemma 3, the above equation implies that

$$\left| \frac{1}{n} \sum_{m=1}^n f(x_m) - \mathbb{E}_x[\mathbb{1}(x(i) > b^*(i) \text{ and } x(j) > b^*(j))] \right| \leq 2\epsilon. \tag{19}$$

By Lemma 2, we have $|\widehat{\theta}_{ij} - \theta_{ij}^*| \leq 4\pi\epsilon$. Lemma 4 follows from the fact that $\cos(\cdot)$ has Lipschitz constant 1. Re-scaling $\epsilon$ gives the desired sample complexity. $\qquad\square$

## D  Proof of Theorem 1

The theorem is restated below, followed by its proof.

**Theorem.** *For any $\epsilon \in (0,1)$ and $\delta \in (0,1)$, Algorithm 1 takes $n = \widetilde{O}\left(\frac{1}{\epsilon^2} \ln(\frac{d}{\delta})\right)$ samples from $\mathcal{D}(W^*, b^*)$ (for some non-negative $b^*$) and outputs $\widehat{\Sigma} \in \mathbb{R}^{d \times d}$ and $\widehat{b} \in \mathbb{R}^d$ that satisfy*

$$\|\widehat{\Sigma} - W^* W^{*T}\|_F \leq \epsilon \|W^*\|_F^2, \quad \|\widehat{b} - b^*\|_2 \leq \epsilon \|W^*\|_F \tag{20}$$

*with probability at least $1 - \delta$. Algorithm 1 runs in time $\widetilde{O}\left(\frac{d^2}{\epsilon^2} \ln(\frac{d}{\delta})\right)$ and space $\widetilde{O}\left(\frac{d}{\epsilon^2} \ln(\frac{d}{\delta}) + d^2\right)$.*

*Proof.* By Lemma 1, the first for-loop of Algorithm 1 needs $\widetilde{O}\left(\frac{1}{\epsilon^2} \ln(\frac{d}{\delta})\right)$ samples and outputs $\widehat{\Sigma}(i,i)$ and $\widehat{b}(i)$ that satisfy for all $i \in [d]$,

$$(1 - \epsilon)\|W^*(i,:)\|_2^2 \leq \widehat{\Sigma}(i,i) \leq (1 + \epsilon)\|W^*(i,:)\|_2^2, \quad |\widehat{b}(i) - b^*(i)| \leq \epsilon \|W^*(i,:)\|_2 \tag{21}$$

with probability at least $1 - \delta$. Since $\epsilon \in (0,1)$, the above equation implies that

$$(1 - \epsilon)\|W^*(i,:)\|_2 \leq \sqrt{\widehat{\Sigma}(i,i)} \leq (1 + \epsilon)\|W^*(i,:)\|_2, \quad \|\widehat{b} - b^*\|_2 \leq \epsilon \|W\|_F. \tag{22}$$

By Lemma 4, if $\widehat{b}$ satisfies (21), then the second for-loop of Algorithm 1 needs $O(\frac{1}{\epsilon^2} \ln(\frac{d^2}{\delta}))$ samples and outputs $\widehat{\theta}_{ij}$ that satisfies

$$|\cos(\widehat{\theta}_{ij}) - \cos(\theta_{ij}^*)| \leq \epsilon, \text{ for all } i \neq j \in [d] \tag{23}$$

with probability at least $1 - \delta$. Combining (22) and (23) gives that for all $i, j \in [d]$,

$$|\widehat{\Sigma}(i,j) - \langle W^*(i,:), W^*(j,:)\rangle| \leq 7\epsilon \|W^*(i,:)\|_2 \|W^*(j,:)\|_2 \tag{24}$$

with probability at least $1 - 2\delta$. To see why (24) is true, suppose (with loss of generality) that $\cos(\theta_{ij}) \geq 0$, then $\widehat{\Sigma}(i, j)$ can be upper bounded by

$$
\begin{aligned}
\widehat{\Sigma}(i,j) &= \sqrt{\widehat{\Sigma}(i,i)\widehat{\Sigma}(j,j)}\cos(\widehat{\theta}_{ij}) \\
&\leq (1+\epsilon)^2 \|W^*(i,:)\|_2 \|W^*(j,:)\|_2 (\cos(\theta_{ij}^*) + \epsilon) \\
&= (1 + 2\epsilon + \epsilon^2) \langle W^*(i,:), W^*(j,:)\rangle + \epsilon(1+\epsilon)^2 \|W^*(i,:)\|_2 \|W^*(j,:)\|_2 \\
&\leq \langle W^*(i,:), W^*(j,:)\rangle + 3\epsilon \langle W^*(i,:), W^*(j,:)\rangle + 4\epsilon\|W^*(i,:)\|_2\|W^*(j,:)\|_2 \\
&\leq \langle W^*(i,:), W^*(j,:)\rangle + 7\epsilon\|W^*(i,:)\|_2\|W^*(j,:)\|_2. 
\end{aligned}
\tag{25}
$$

The lower bound can be derived in a similar way. Given (24), we can bound $\|\widehat{\Sigma} - W^*W^{*T}\|_F$ as

$$
\begin{aligned}
\|\widehat{\Sigma} - W^*W^{*T}\|_F^2 &= \sum_{i,j\in[d]} \left( \widehat{\Sigma}(i,j) - \langle W^*(i,:), W^*(j,:)\rangle \right)^2 \\
&\leq \sum_{i,j\in[d]} 49\epsilon^2 \|W^*(i,:)\|_2^2 \|W^*(j,:)\|_2^2 \\
&\leq 49\epsilon^2 \|W\|_F^2 \sum_{i\in[d]} \|W^*(i,:)\|_2^2 = 49\epsilon^2 \|W^*\|_F^4,
\end{aligned}
\tag{26}
$$

which holds with probability at least $1 - 2\delta$. Re-scaling $\epsilon$ and $\delta$ gives the desired bound in Theorem 1. The final sample complexity is $\widetilde{O}\left(\frac{1}{\epsilon^2}\ln(\frac{d}{\delta})\right) + O(\frac{1}{\epsilon^2}\ln(\frac{d^2}{\delta})) = \widetilde{O}\left(\frac{1}{\epsilon^2}\ln(\frac{d}{\delta})\right)$.

We now analyze the time complexity. The first for-loop runs in time $O(dn)$, where $n$ is the number of input samples. Note that in Step 3 of Algorithm 3, gradient estimation requires sampling from a truncated normal distribution. This can be done by sampling from a normal distribution until it falls into the truncation set. The probability of hitting a truncation set is lower bounded by a constant (Lemma 7 of [DGTZ18]). The second for-loop of Algorithm 1 runs in time $O(d^2n)$. The space complexity is determined by the space required to store $n$ samples and the matrix $\widehat{\Sigma} \in \mathbb{R}^{d\times d}$, which is $O(dn + d^2)$.  $\qquad\square$

## E  Proof of Corollary 1

We first restate the corollary and then give the proof.

**Corollary.** *Suppose that $W^* \in \mathbb{R}^{d\times d}$ is full-rank. Let $\kappa$ be the condition number of $W^*W^{*T}$. For any $\epsilon \in (0, 1/2]$ and $\delta \in (0, 1)$, Algorithm 1 takes $n = \widetilde{O}\left(\frac{\kappa^2 d^2}{\epsilon^2}\ln(\frac{d}{\delta})\right)$ samples from $\mathcal{D}(W^*, b^*)$ (for some non-negative $b^*$) and outputs a distribution $\mathcal{D}(\widehat{\Sigma}^{1/2}, \widehat{b})$ that satisfies*

$$
\mathrm{TV}\left( \mathcal{D}(\widehat{\Sigma}^{1/2}, \widehat{b}),\ \mathcal{D}(W^*, b^*) \right) \leq \epsilon,
$$

*with probability at least $1 - \delta$. Algorithm 1 runs in time $\widetilde{O}\left(\frac{\kappa^2 d^4}{\epsilon^2}\ln(\frac{d}{\delta})\right)$ and space $\widetilde{O}\left(\frac{\kappa^2 d^3}{\epsilon^2}\ln(\frac{d}{\delta})\right)$.*

*Proof.* Let $\Sigma = W^*W^{*T}$. We will prove that given $\widetilde{O}\left(\frac{\kappa^2 d^2}{\epsilon^2}\ln(\frac{d}{\delta})\right)$ samples, the output of Algorithm 1 satisfies

$$
\|\Sigma^{-1/2}(\widehat{b} - b^*)\|_2 \leq \epsilon, \quad \|\Sigma^{-1/2}\widehat{\Sigma}\Sigma^{-1/2} - I\|_F \leq \epsilon.
\tag{27}
$$

The above implies that the TV distance between $\mathcal{D}(\widehat{\Sigma}^{1/2}, \widehat{b})$ and $\mathcal{D}(W^*, b^*)$ is less than $\epsilon$. To see why, note that

$$
\mathrm{TV}\left( \mathcal{D}(\widehat{\Sigma}^{1/2}, \widehat{b}),\ \mathcal{D}(W^*, b^*) \right) \leq \mathrm{TV}\left( \mathcal{N}(\widehat{b}, \widehat{\Sigma}), \mathcal{N}(b^*, \Sigma) \right)
$$

$$
\leq \sqrt{\mathrm{KL}\left( \mathcal{N}(\widehat{b}, \widehat{\Sigma}) \| \mathcal{N}(b^*, \Sigma) \right)/2}.
\tag{28}
$$

The first inequality follows from the data processing inequality for $f$-divergence given by Lemma 7 in Appendix F (see also [ABDH$^+$18, Fact A.5]): $\mathrm{TV}(f(X), f(Y)) \leq \mathrm{TV}(X, Y)$ for any function $f$

and random variables $X, Y$ over the same space. The second inequality follows from the Pinsker's inequality [Tsy09, Lemma 2.5]. The KL divergence between two Gaussian distributions can be computed as KL $\left( \mathcal{N}(\widehat{b}, \widehat{\Sigma}) \| \mathcal{N}(b^*, \Sigma) \right) =$

$$\frac{1}{2} \left( \text{tr}(\Sigma^{-1}\widehat{\Sigma} - I) - \ln(\det(\Sigma^{-1}\widehat{\Sigma})) + \|\Sigma^{-1/2}(\widehat{b} - b^*)\|_2^2 \right). \tag{29}$$

Let $\lambda_1, ..., \lambda_d$ be the eigenvalues of $\Sigma^{-1}\widehat{\Sigma}$. We have

$$\text{tr}(\Sigma^{-1}\widehat{\Sigma} - I) - \ln(\det(\Sigma^{-1}\widehat{\Sigma})) = \sum_{i=1}^{d}(\lambda_i - 1) - \ln(\Pi_{i=1}^{d}\lambda_i) = \sum_{i=1}^{d}(\lambda_i - 1 - \ln(\lambda_i)). \tag{30}$$

Suppose that (27) holds with $\epsilon \leq 1/2$, since $\Sigma^{-1/2}\widehat{\Sigma}\Sigma^{-1/2}$ and $\Sigma^{-1}\widehat{\Sigma}$ have the same eigenvalues,

$$\epsilon^2 \geq \|\Sigma^{-1/2}\widehat{\Sigma}\Sigma^{-1/2} - I\|_F^2 = \sum_{i=1}^{d}(\lambda_i - 1)^2 \geq \sum_{i=1}^{d}(\lambda_i - 1 - \ln(\lambda)), \tag{31}$$

where the last inequality follows from the fact that $x - 1 - \ln(x) \leq (x-1)^2$ for $x \geq 1/2$. Since $\epsilon \leq 1/2$, we have $(\lambda_i - 1)^2 \leq 1/4$, which implies that $\lambda_i \in [1/2, 3/2]$. Substituting (31) into (30), and combining (29) and (28) give that the TV $\left( \mathcal{D}(\widehat{\Sigma}^{1/2}, \widehat{b}), \ \mathcal{D}(W^*, b^*) \right) \leq \epsilon$.

The only thing left is to prove that (27) holds. According to Theorem 1, given $\widetilde{O}\left( \frac{1}{\eta^2}\ln(\frac{d}{\delta}) \right)$ samples, we have

$$\|\widehat{\Sigma} - \Sigma\|_F \leq \eta\|W^*\|_F^2, \quad \|\widehat{b} - b^*\|_2 \leq \eta\|W^*\|_F. \tag{32}$$

We can bound $\|\Sigma^{-1/2}(\widehat{b} - b^*)\|_2$ and $\|\Sigma^{-1/2}\widehat{\Sigma}\Sigma^{-1/2} - I\|_F$ as

$$\|\Sigma^{-1/2}(\widehat{b} - b^*)\|_2 \leq \|\Sigma^{-1/2}\|_2\|\widehat{b} - b\|_2 \leq \|\Sigma^{-1/2}\|_2\eta\|W^*\|_F \leq \eta\sqrt{\kappa d}.$$

$$\|\Sigma^{-1/2}\widehat{\Sigma}\Sigma^{-1/2} - I\|_F = \|\Sigma^{-1/2}(\widehat{\Sigma} - \Sigma)\Sigma^{-1/2}\|_F \leq \|\Sigma^{-1/2}\|_2^2\|\widehat{\Sigma} - \Sigma\|_F \leq \eta\kappa d.$$

Now setting $\eta = \epsilon/(\kappa d)$ gives (27). $\qquad\qquad\qquad\qquad\qquad\qquad\qquad\qquad\qquad\qquad\qquad\square$

# F   Proof of Theorem 2

To establish a lower bound for parameter estimation, the key step is to construct a local packing set such that their parameter distance is large but their KL divergence is small (and hence it is hard to distinguish them without observing many samples). We remark that our way of constructing this local packing is similar to the one used in proving the minimax rate for Gaussian mean estimation (see, e.g., [Duc19]), despite the fact that our class of distributions is not Gaussian.

We will start by stating three results in information theory and statistics. Proofs of Lemma 5, 6, and 7 can be found in, e.g., [Duc19].

**Lemma 5.** *(Gilbert-Varshamov bound). There is a subset $\mathcal{V}$ of the $d$-dimensional hypercube $\{0, 1\}^d$ of size $|\mathcal{V}| \geq \exp(d/8)$ such that the $\ell_1$-distance*

$$\|v - v'\|_1 = \sum_{j=1}^{d} \mathbb{1}(v_j \neq v'_j) \geq d/4, \quad \text{for any } v, v' \in \mathcal{V}. \tag{33}$$

**Lemma 6.** *(Fano's inequality). Let $V$ be a random variable taking values uniformly in the finite set $\mathcal{V}$ with cardinality $|\mathcal{V}| \geq 2$. Conditioned on $V = v$, we draw a sample $X \sim P_v$. The KL divergence of the distributions $\{P_v\}_{v \in \mathcal{V}}$ satisfy*

$$\text{KL}(P_v \| P_{v'}) \leq \beta, \quad \text{for any } v, v' \in \mathcal{V}. \tag{34}$$

*For any Markov chain $V \to X \to \widehat{V}$,*

$$\mathbb{P}[\widehat{V} \neq V] \geq 1 - \frac{\beta + \ln(2)}{\ln(|\mathcal{V}|)}. \tag{35}$$

**Lemma 7.** *(Data processing inequality for $f$-divergence). Let $f_1$ and $f_2$ be the distributions of two random variables $x_1$ and $x_2$. Let $g_1$ and $g_2$ be the distributions of two random variables $T(x_1)$ and $T(x_2)$, where $T(\cdot)$ is any function. For any $f$-divergence $D_f(\cdot \parallel \cdot)$, we have*

$$D_f(f_1 \parallel f_2) \geq D_f(g_1 \parallel g_2). \tag{36}$$

We are now ready to prove Theorem 2, which is restated below.

**Theorem.** *Let $\sigma > 0$ be a fixed and known scalar. Let $I_d$ be the identity matrix in $\mathbb{R}^d$. Let $S := \{\mathcal{D}(W, b) : W = \sigma I_d, b \in \mathbb{R}^d$ non-negative$\}$ be a class of distributions in $\mathbb{R}^d$. Any algorithm that learns $S$ to satisfy $\|\widehat{b} - b^*\|_2 \leq \epsilon \|W^*\|_F$ with success probability at least 2/3 requires $\Omega(\frac{1}{\epsilon^2})$ samples.*

*Proof.* Let $\mathcal{V} \subset \{0, 1\}^d$ be a finite set satisfying the property in Lemma 5. Given an $\epsilon \in (0, 1)$, we can construct a finite set of distributions $\{P_v\}_{v \in V}$ as follows:

$$P_v = \mathcal{D}(\sigma I_d, b_v), \text{ where } b_v = 6\epsilon \sigma v. \tag{37}$$

Clearly $\{P_v\}_{v \in V}$ belong to the class of the distributions that we are interested in. Furthermore, they satisfy two properties:

- Property 1: $\|b_v - b_{v'}\|_2 \geq 3\epsilon \sigma \sqrt{d}$ and $|\mathcal{V}| \geq \exp(d/8)$.

- Property 2: $\mathrm{KL}(P_v \parallel P_{v'}) \leq 18 d\epsilon^2$.

Assuming that the above two properties hold, we can use Fano's inequality (Lemma 6) to obtain a sample complexity lower bound for learning $\{P_v\}_{v \in \mathcal{V}}$. Let $V$ be a random variable taking values uniformly in $\mathcal{V}$. Conditioned on $V = v$, we draw $n$ i.i.d. samples $X^n \sim P_v^n$, where $P_v^n$ represents a product distribution of $n$ $P_v$'s. Given $X^n$, our goal is to output an index $\widehat{v} \in \mathcal{V}$. By Lemma 6, any estimator will suffer an estimation error larger than

$$\mathbb{P}[\widehat{V} \neq V] \geq 1 - \frac{18 n d\epsilon^2 + \ln(2)}{d/8}, \tag{38}$$

which follows from the fact that $|\mathcal{V}| \geq \exp(d/8)$ (Property 1) and $\mathrm{KL}(P_v^n \| P_{v'}^n) = n\mathrm{KL}(P_v \| P_{v'}) \leq 18 n d\epsilon^2$ (Property 2). Eq. (38) implies that any estimator that estimates the index correctly with probability at least 2/3 must observe $\Omega(\frac{1}{\epsilon^2})$ samples. Furthermore, by Property 1, $\|b_v - b_{v'}\|_2 \geq 3\epsilon \sigma \sqrt{d}$, any algorithm that learns $S$ to satisfy $\|\widehat{b} - b^*\|_2 \leq \epsilon \|W^*\|_F = \epsilon \sigma \sqrt{d}$ can be used to estimate $\mathcal{V}$ (we can just choose $\widehat{v} \in \mathcal{V}$ such that $b_{\widehat{v}}$ is closest to $\widehat{b}$). Therefore, any algorithm that learns $S$ to satisfy $\|\widehat{b} - b^*\|_2 \leq \epsilon \|W^*\|_F$ with success probability at least 2/3 requires $\Omega(\frac{1}{\epsilon^2})$ samples.

The only thing left is to show that Property 1 and 2 hold. Property 1 follows from Lemma 5 and the way we construct $P_v$. Property 2 is true because of the following two facts.

- Fact 1: The KL-divergence between two Gaussian distributions can be computed as

$$\mathrm{KL}(\mathcal{N}(b_v, \sigma^2 I_d) \parallel \mathcal{N}(b_{v'}, \sigma^2 I_d)) = \frac{\|b_v - b_{v'}\|_2^2}{2\sigma^2} = 18 d\epsilon^2. \tag{39}$$

- Fact 2: $\mathrm{KL}(P_v \parallel P_{v'}) \leq \mathrm{KL}(\mathcal{N}(b_v, \sigma^2 I_d) \parallel \mathcal{N}(b_{v'}, \sigma^2 I_d))$, which follows from Lemma 7 and the fact that KL-divergence is an instance of $f$-divergence.

$\square$

# G  Proof of Theorem 3

We first restate the theorem, and then give the proof.

**Theorem.** *Let $S := \{\mathcal{D}(W, 0) : W \in \mathbb{R}^{d \times d}$ full rank$\}$ be a set of distributions in $\mathbb{R}^d$. Any algorithm that learns $S$ within total variation distance $\epsilon$ and success probability at least 2/3 requires $\Omega(\frac{d}{\epsilon^2})$ samples.*

*Proof.* Similar to the proof of Theorem 2, we construct a local packing of $S$ for which their pairwise TV distance is large while their KL-divergence is small. Let $\mathcal{V} \subset \{0,1\}^d$ be a finite set satisfying the property in Lemma 5. Given an $\epsilon \in (0,1)$, define $\lambda = C \cdot \epsilon/\sqrt{d}$, where $C$ is a universal constant to be specified later, we can construct a finite set of distributions $\{P_v\}_{v \in V}$ as follows:

$$P_v = \mathcal{D}(W_v, 0), \text{ where } W_v = I_d + \lambda \cdot \text{diag}(v). \tag{40}$$

Here $\text{diag}(\cdot) : \mathbb{R}^d \to \mathbb{R}^{d \times d}$ defines a diagonal matrix. This finite set of distributions satisfies two properties:

- Property 1: $\text{TV}(P_v, P_{v'}) \geq 3\epsilon$ and $|\mathcal{V}| \geq \exp(d/8)$.

- Property 2: $\text{KL}(P_v \parallel P_{v'}) = O(\epsilon^2)$.

Given the above two properties, we can use Fano's inequality (Lemma 6) in a way similar to the proof of Theorem 2 to conclude that any estimator that identifies $P_v$ from i.i.d. samples with success probability at least 2/3 must require $\Omega(d/\epsilon^2)$ samples. Since $\text{TV}(P_v, P_{v'}) \geq 3\epsilon$, any algorithm that learns $S$ within TV distance $\epsilon$ can be used to estimate $\{P_v\}_{v \in \mathcal{V}}$ (we can just choose $P_v$ that has the smallest TV distance to the output of the algorithm). This implies that any algorithm that learns $S$ within TV distance $\epsilon$ with success probability at least 2/3 requires $\Omega(d/\epsilon^2)$ samples.

The only thing left is to show that the two properties hold for our packing set $\{P_v\}_{v \in \mathcal{V}}$. To prove Property 2, note that

$$\text{KL}(P_v \parallel P_{v'}) \overset{(a)}{\leq} \text{KL}(\mathcal{N}(0, W_v W_v^T) \parallel \mathcal{N}(0, W_{v'} W_{v'}^T)) \overset{(b)}{=} O(\lambda^2 d) = O(\epsilon^2), \tag{41}$$

where (a) follows from Lemma 7 and the fact that KL-divergence belongs to $f$-divergence; (b) follows from exactly computing the KL-divergence between the two Gaussian distributions. Before computing that, we need a few more notations. Specifically, let $S_v = \{i \in [d] : W_v(i,i) = 1 + \lambda\}$ be the set of coordinates that the corresponding diagonal entry of $W_v$ is $1 + \lambda$. We use $S_v - S_{v'} = \{i \in S_v : i \neq S_{v'}\}$ to denote the difference between two sets. For simplicity, we write $\Sigma_v = W_v W_v^T$. Now we can compute the KL-divergence between the two Gaussian distributions as

$$2\text{KL}(\mathcal{N}(0, W_v W_v^T) \parallel \mathcal{N}(0, W_{v'} W_{v'}^T))$$

$$= \text{Tr}\left(\Sigma_{v'}^{-1} \Sigma_v - I_d\right) + \ln\left(\det(\Sigma_{v'})\right) - \ln\left(\det(\Sigma_v)\right)$$

$$= |S_v - S_{v'}|\left((1+\lambda)^2 - 1\right) + |S_{v'} - S_v|\left(\frac{1}{(1+\lambda)^2} - 1\right)$$

$$\quad + 2|S_{v'}| \ln(1+\lambda) - 2|S_v| \ln(1+\lambda)$$

$$\overset{(a)}{\leq} |S_v|\left[(1+\lambda)^2 - 1 - 2\ln(1+\lambda)\right] + |S_{v'}|\left[2\ln(1+\lambda) + \frac{1}{(1+\lambda)^2} - 1\right]$$

$$\overset{(b)}{\leq} |S_v|\left(2\lambda + \lambda^2 - \frac{2\lambda}{1+\lambda}\right) + |S_{v'}|\left(2\lambda - \frac{2\lambda + \lambda^2}{(1+\lambda)^2}\right)$$

$$= |S_v|\frac{3\lambda^2 + \lambda^3}{1+\lambda} + |S_{v'}|\frac{3\lambda^2 + 2\lambda^3}{(1+\lambda)^2}$$

$$\overset{(c)}{=} O(d\lambda^2) = O(\epsilon^2),$$

where (a) follows from $|S_v| \leq |S_v - S_{v'}|$, (b) follows from $\ln(1+x) \leq x$, and (c) is true because $|S_v| \leq d$ and $\lambda \in (0,1)$. Substituting $\lambda = O(\epsilon/\sqrt{d})$ gives the final result.

To prove Property 1, note that $|\mathcal{V}| \leq \exp(d/8)$ directly follows from Lemma 5. The key challenge lies in proving a lower bound for $\text{TV}(P_v, P_{v'})$. Note that the data-processing inequality (i.e., Lemma 7) only implies that $\text{TV}(P_v, P_{v'}) \leq \text{TV}(\mathcal{N}(0, W_v W_v^T), \mathcal{N}(0, W_{v'} W_{v'}^T))$, so we cannot use the TV distance for Gaussian to obtain a lower bound on the TV distance for rectified Gaussian. Our proof strategy instead is to directly compute the TV distance for the specially-constructed $\{P_v\}_{v \in \mathcal{V}}$ (computing the exact TV distance is hard for general rectified Gaussian distributions). Specifically, let $\Sigma_v = W_v W_v^T$, our proof uses the following two facts:

- Fact 1: $\text{TV}(\mathcal{N}(0, \Sigma_v), \mathcal{N}(0, \Sigma_{v'})) \geq 0.01 \|\Sigma_v^{-1} \Sigma_{v'} - I_d\|_F \geq C' \cdot \lambda\sqrt{d}$, where $C'$ is a universal constant.

- Fact 2: Let $Q_v$ be the probability density function of a multivariate normal distribution $\mathcal{N}(0, \Sigma_v)$. Let $\mathbb{R}_{>0}^d = \{x \in \mathbb{R}^d : x > 0 \text{ coordinate-wise}\}$ be the (open) positive orthant. Then

$$\|Q_v - Q_{v'}\|_1 = \int_{\mathbb{R}^d} |Q_v(x) - Q_{v'}(x)| \, \mathrm{d}x = 2^d \int_{\mathbb{R}_{>0}^d} |Q_v(x) - Q_{v'}(x)| \, \mathrm{d}x.$$

The first inequality in Fact 1 follows from [DMR18, Theorem 1.1]. The second inequality follows from our definition of $\Sigma_v$. Specifically, the diagonal entry of $\Sigma_v$ is either 1 or $1 + \lambda$. By Lemma 5, we know that $\Sigma_v$ and $\Sigma_{v'}$ have at least $d/4$ different diagonal entries. Since the total variation distance is symmetric, i.e., $\mathrm{TV}(\mathcal{N}(0, \Sigma_v), \mathcal{N}(0, \Sigma_{v'})) = \mathrm{TV}(\mathcal{N}(0, \Sigma_{v'}), \mathcal{N}(0, \Sigma_v))$, we can w.l.o.g assume that among the diagonal entries that $\Sigma_v$ is different from $\Sigma_{v'}$, $\Sigma_{v'}$ has more entries with value $1 + \lambda$ than entries with value 1. This then implies that $\|\Sigma_v^{-1}\Sigma_{v'} - I_d\|_F = \Omega(\lambda\sqrt{d})$.

Fact 2 is true because $\mathcal{N}(0, \Sigma_v)$ has zero mean and diagonal covariance matrix, and hence the value of $Q_v(x)$ is invariant to the sign of $x$'s coordinates.

Now we prove a lower bound on $\mathrm{TV}(P_v, P_{v'})$, assuming that are all the $d$ diagonal entries of $\Sigma_v$ and $\Sigma_{v'}$ are different. Let $\Omega \subseteq [d]$ be any subset of the $d$ coordinates. For any $\Omega$, let $x_\Omega \in \mathbb{R}^{|\Omega|}$ be the sub-vector of $x \in \mathbb{R}^d$ over the coordinates in $\Omega$. Let $\Omega^c = [d] - \Omega$ be its complement. We can re-write $\mathrm{TV}(P_v, P_{v'})$ as a summation of integrals, where each integral is over the space $A_\Omega = \{x \in \mathbb{R}^d : x_\Omega > 0, \ x_{\Omega^c} = 0\}$:

$$\mathrm{TV}(P_v, P_{v'}) = \|P_v - P_{v'}\|_1 = \sum_\Omega \int_{x \in A_\Omega} |P_v(x) - P_{v'}(x)| \, \mathrm{d}x. \tag{42}$$

We now give a lower bound for every integral. Let $\Sigma_{v,\Omega} \in \mathbb{R}^{|\Omega| \times |\Omega|}$ be the sub-matrix of $\Sigma_v$ over the coordinates in $\Omega$. Since $\Sigma_v$ has zero mean and diagonal covariance matrix, for any $\Omega \subseteq [d]$ and any $x \in A_\Omega$, we have $P_v(x) = (\frac{1}{2})^{|\Omega^c|} P_{v,\Omega}(x_\Omega)$, where $P_{v,\Omega}$ is the probability density function of the normal distribution $\mathcal{N}(0, \Sigma_{v,\Omega})$. By Fact 1 and 2, we have

$$\int_{x \in A_\Omega} |P_v(x) - P_{v'}(x)| \, \mathrm{d}x = (\frac{1}{2})^{|\Omega^c|} \cdot \frac{1}{2^{|\Omega|}} \mathrm{TV}(\mathcal{N}(0, \Sigma_{v,\Omega}), \mathcal{N}(0, \Sigma_{v',\Omega}))$$
$$\geq \frac{C' \cdot \lambda \sqrt{|\Omega|}}{2^d}. \tag{43}$$

Combining (42) and (43) gives

$$\mathrm{TV}(P_v, P_{v'}) \geq \sum_{i=0}^d \binom{d}{i} \frac{C' \cdot \lambda\sqrt{i}}{2^d}$$
$$\geq \sum_{i=\lfloor d/2 \rfloor}^d \binom{d}{i} \frac{C' \cdot \lambda\sqrt{\lfloor d/2 \rfloor}}{2^d}$$
$$\overset{(a)}{\geq} \frac{C' \cdot \lambda\sqrt{\lfloor d/2 \rfloor}}{2^d} \frac{1}{2} \sum_{i=0}^d \binom{d}{i}$$
$$\overset{(b)}{=} \frac{C' \cdot \lambda\sqrt{\lfloor d/2 \rfloor}}{2} \overset{(c)}{=} 3\epsilon, \tag{44}$$

where (a) follows from the fact that $\binom{d}{i} = \binom{d}{d-i}$, (b) is true because $\sum_i \binom{d}{i} = 2^d$, and (c) holds if we choose $\lambda = C \cdot \epsilon/\sqrt{d}$ with a proper constant $C$.

So far we have proved that $\mathrm{TV}(P_v, P_{v'}) \geq 3\epsilon$ when all the $d$ diagonal entries of $\Sigma_v$ and $\Sigma_{v'}$ are different. The proof can be easily extended when only a subset of their diagonal entries are different. Let $\Omega \subset [d]$ be the subset of $d$ diagonal entries that $\Sigma_v$ and $\Sigma_{v'}$ are different. By Lemma 5, we know

that $|\Omega| \geq d/4$. The definition of TV distance gives

$$
\begin{aligned}
\mathrm{TV}(P_v, P_{v'}) &= \int_x |P_v(x) - P_{v'}(x)| \, \mathrm{d}x \\
&\stackrel{(a)}{=} \int_{x^{\Omega^c}} \int_{x^{\Omega}} |P_{v^{\Omega}}(x^{\Omega}) P_{v^{\Omega^c}}(x^{\Omega^c}) - P_{(v')^{\Omega}}(x^{\Omega}) P_{(v')^{\Omega^c}}(x^{\Omega^c})| \, \mathrm{d}x^{\Omega} \, \mathrm{d}x^{\Omega^c} \\
&\stackrel{(b)}{=} \int_{x^{\Omega}} |P_{v^{\Omega}}(x^{\Omega}) - P_{(v')^{\Omega}}(x^{\Omega})| \, \mathrm{d}x^{\Omega} \int_{x^{\Omega^c}} P_{v^{\Omega^c}}(x^{\Omega^c}) \, \mathrm{d}x^{\Omega^c} \\
&= \int_{x^{\Omega}} |P_{v^{\Omega}}(x^{\Omega}) - P_{(v')^{\Omega}}(x^{\Omega})| \, \mathrm{d}x^{\Omega} \\
&= \mathrm{TV}(P_{v^{\Omega}}, P_{(v')^{\Omega}}).
\end{aligned}
\tag{45}
$$

Here equality (a) uses the fact that $P_v$ and $P_{v'}$ have independent coordinates as $\Sigma_v$ and $\Sigma_{v'}$ are diagonal matrices. Equality (b) follows from the definition of $\Omega$: the diagonal entries in $\Omega^c$ are the same for $\Sigma_v$ and $\Sigma_{v'}$, and hence, $P_{v^{\Omega^c}}(x^{\Omega^c}) = P_{(v')^{\Omega^c}}(x^{\Omega^c})$.

By (45), we have proved that the TV distance between $P_v$ and $P_{v'}$ equals the TV distance between the two distributions over the coordinates in $\Omega$. By definition, $\Sigma_{v^{\Omega}} \in \mathbb{R}^{|\Omega| \times |\Omega|}$ and $\Sigma_{(v')^{\Omega}} \in \mathbb{R}^{|\Omega| \times |\Omega|}$ have different diagonal entries, and $|\Omega| \geq d/4$, we can use the same proof in (44) to show that $\mathrm{TV}(P_{v^{\Omega}}, P_{(v')^{\Omega}}) \geq 3\epsilon$ for small enough $\lambda$. $\qquad\square$

## H  Open Problems

### H.1  Negative Bias

Our algorithm relies on the assumption that the bias vector is non-negative. This assumption is required to ensure that Lemma 2 holds, which subsequently ensures that the pairwise angles between the row vectors of $W^*$ can be correctly recovered. A weaker assumption would be allowing the bias vector $b^*$ to be negative but constraining the largest negative values. Designing algorithms under this weaker assumption is an interesting direction for future research.

When $b^*$ has negative components, running our algorithm can still recover part of the parameters with a small number of samples. Specifically, let $\Omega := \{i \in [d] : b^*(i) \geq 0\}$ be the set of coordinates that $b^*$ is non-negative; let $b^*_{\Omega} \in \mathbb{R}^{|\Omega|}$ and $W^*_{\Omega} \in \mathbb{R}^{|\Omega| \times k}$ be the sub-vector and sub-matrix associated with the coordinates in $\Omega$. Then given $O(\frac{1}{\epsilon^2} \ln(\frac{d}{\delta}))$ samples, the output of our algorithm $\widehat{b} \in \mathbb{R}^d$ and $\widehat{\Sigma} \in \mathbb{R}^{d \times d}$ satisfies

$$
\|\widehat{\Sigma}_{\Omega \times \Omega} - W^*_{\Omega} W^{*T}_{\Omega}\|_F \leq \epsilon \|W^*_{\Omega}\|_F^2, \quad \|\widehat{b}_{\Omega} - b^*_{\Omega}\|_2 \leq \epsilon \|W^*_{\Omega}\|_F,
$$

with probability at least $1 - \delta$. The above guarantee is the same as Theorem 1. The reason is that our algorithm only uses the $i$-th and $j$-th coordinates of the samples to estimate $\langle W^*(i, :), W^*(j, :) \rangle$ and $b^*(i), b^*(j)$. As a result, Theorem 1 still holds for this part of the parameters.

For the rest part of the parameters, if the negative components of $b^*$ are small (in absolute value), then the error of our algorithm will be also small. Let $\Omega^c$ be the complement of $\Omega$. We assume that there is a value $\eta \geq 0$ such that the negative coordinates of $b^*$ satisfy

$$
b^*(i) \geq -\eta \|W^*(i, :)\|_2, \quad \text{for all } i \in \Omega^c.
$$

Given $\widetilde{O}(\ln(d)/\epsilon^2)$ samples, the output of our algorithm satisfies

$$
|\widehat{b}(i) - b^*(i)| \leq \max(\eta, \epsilon) \|W^*(i, :)\|_2, \quad \text{for all } i \in \Omega^c.
$$

One can show a similar result for $\langle W^*(i, :), W^*(j, :) \rangle$, where $i \in \Omega^c$ and $j \in [d]$:

$$
|\widehat{\Sigma}(i, j) - \langle W^*(i, :), W^*(j, :) \rangle| \leq 7 \max(\eta, \epsilon) \|W^*(i, :)\|_2 \|W^*(j, :)\|_2.
$$

Comparing the above two equations with (21) and (24), we see that the error from the negative bias is small if $\eta = O(\epsilon)$. If $\eta$ is large, i.e., if $b^*$ have large negative components, then estimating those parameters becomes difficult (as indicated by Claim 2). In that case, maybe one should directly estimate the distribution without estimating the parameters. This is an interesting direction for future research.

## H.2 Two-Layer Generative Model

One natural generalization of our problem is to consider distributions defined by a two-layer generative model:

**Definition 3.** *Given $A \in \mathbb{R}^{d \times p}$, $W \in \mathbb{R}^{p \times k}$, and $b \in \mathbb{R}^p$, we define $\mathcal{D}(A, W, b)$ as the distribution of a random variable $x \in \mathbb{R}^d$ generated as follows:*

$$x = A \operatorname{ReLU}(Wz + b), \text{ where } z \sim \mathcal{N}(0, I_k). \tag{46}$$

Given i.i.d. samples $x \sim \mathcal{D}(A, W, b)$, can we recover the parameters $A, W, b$ (up to permutation and scaling of the columns of $A$)? While this problem seems hard in general, we find an interesting connection between this problem and non-negative matrix factorization. A non-negative matrix has all its entries being non-negative. Note that in our problem, the $A$ matrix does *not* need to be a non-negative matrix.

**Connection to Non-negative Matrix Factorization (NMF).** In MNF, we are given a non-negative matrix $X \in \mathbb{R}^{d \times n}$ and an integer $p > 0$, the goal is to find two non-negative matrices $A \in \mathbb{R}^{d \times p}, M \in \mathbb{R}^{p \times n}$ such that $X = AM$. This problem is NP-hard in general [Vav09]. Arora et al. [AGKM12] give the first polynomial-time algorithm under the "separability" condition [DS04]:

**Definition 4.** *The factorization $X = AM$ is called separable[9] if for each $i \in [p]$, there is a column $f(i) \in [n]$ of $M$ such that $M(:, f(i)) \in \mathbb{R}^p$ has only one non-zero positive entry at the $i$-th location, i.e., $M(i, f(i)) > 0$ and $M(j, f(i)) = 0$ for $j \neq i$.*

If the separability condition holds, then the algorithm proposed in [AGKM12] is guaranteed to find a separable non-negative factorization in time polynomial in $n, p, d$.

In our problem, we are given $n$ samples $\{x_i\}_{i=1}^n$ from $\mathcal{D}(A, W, b)$. Stacking these samples to form a matrix $X \in \mathbb{R}^{d \times n}$ as

$$X = AM, \text{ where } M(:, i) = \operatorname{ReLU}(Wz_i + b), i \in [n]. \tag{47}$$

Note that $M \in \mathbb{R}^{p \times n}$ is a non-negative matrix while $A$ can be an arbitrary matrix. Nevertheless, if $M$ satisfies the separability condition (Definition 4), and $A$ has full column rank (i.e., the columns of $A$ are linearly independent), then we can still use the same idea of [AGKM12] to *exactly* recover $A$ and $M$ (up to permutation and scaling of the column vectors in $A$). Once $M \in \mathbb{R}^{p \times n}$ is recovered, estimating $W$ and $b$ is the same problem as learning one-layer ReLU generative model, and hence can be done by our algorithm (Algorithm 1) assuming that $b$ is non-negative.

The pseudocode is given in Algorithm 4. We first create a set $S$ by normalizing each sample and removing zero and duplicated vectors. The next step is to check for each vector $v \in S$, whether $v$ can be represented as a conical sum (i.e., non-negative linear combination) of the rest vectors in $S$. This can be done by checking the feasibility of a linear program. For example, checking whether vector $v$ can be expressed as a conical sum of two vectors $w_1, w_2$ is equivalent to checking whether the following linear program is feasible:

$$\min_{c_1 \geq 0, \, c_2 \geq 0} c_1 + c_2 \quad \text{s.t. } c_1 w_1 + c_2 w_2 = v.$$

We only keep a vector if it *cannot* be written as the conical sum of the other vectors. Those vectors are then stacked to form $\widehat{A}$. Let $\widehat{A}^\dagger = (\widehat{A}^T \widehat{A})^{-1} \widehat{A}^T$ be the pseudo-inverse of $\widehat{A}$. The last step is to compute $\{\widehat{A}^\dagger x_i\}_{i=1}^n$ and treat them as samples from one-layer ReLU generative model so that we can run Algorithm 1 to estimate $W^* W^{*T}$ and $b^*$.

**Claim 3.** *Define $X \in \mathbb{R}^{d \times n}$ and $M \in \mathbb{R}^{p \times n}$ as in (47). Without loss of generality, we assume that the column vectors of $A^*$ have unit $\ell_2$-norm. Let $\widehat{A}$ be the output of Algorithm 4. If $A^*$ has full column rank, and $M$ satisfies the separability condition in Definition 4, then there is a way to permute the column vectors of $\widehat{A}$ so that $\widehat{A} = A^*$.*

*Proof.* After Step 1-7, Algorithm 4 produces a set $S$ which contains all nonzero and normalized samples. Besides, the vectors in $S$ are unique because the duplicated ones are removed in Step 7. To prove $\widehat{A} = A^*$ (up to permutation of the columns), we only need to prove that

**Algorithm 4:** Learning a two-layer ReLU generative model

---

**Input:** $n$ i.i.d. samples $x_1, \cdots, x_n \in \mathbb{R}^d$ from $\mathcal{D}(A^*, W^*, b^*)$, $b^*$ is non-negative, $A^*$ has linearly independent column vectors.

    **Output:** $\widehat{A} \in \mathbb{R}^{d \times p}$, $\widehat{\Sigma} \in \mathbb{R}^{p \times p}$, $\widehat{b} \in \mathbb{R}^p$.

**1** $S \leftarrow \emptyset$;

**2 for** $i \leftarrow 1$ **to** $n$ **do**

**3**     **if** $x_i \neq 0$ **then**

**4**         | $S \leftarrow S \cup \{x_i / \|x_i\|_2\}$;

**5**     **end**

**6 end**

**7** Remove duplicated vectors from $S$;

**8 for** $v \in S$ **do**

**9**     **if** $v$ is a conical sum of the rest vectors in $S$ **then**

**10**         | Remove $v$ from $S$;

**11**     **end**

**12 end**

**13** $\widehat{A} \leftarrow$ stack vectors from $S$;

**14** $\widehat{\Sigma}, \widehat{b} \leftarrow$ run Algorithm 1 with samples $\{\widehat{A}^\dagger x_i\}_{i=1}^n$.

---

- (a) All the (normalized) column vectors of $A^*$ are in $S$.

- (b) Except the column vectors in $A^*$, every vector in $S$ can be represented as a conical sum of the rest vectors in $S$.

- (c) Any column vector in $A^*$ cannot be represented as a conical sum of the rest vectors in $S$.

(a) is true because the $M$ matrix satisfies the separability condition. According to Definition 4, for every column vector of $A^*$, there is at least one sample $x \in \mathbb{R}^d$ which is a scaled version of that column vector.

To prove (b), first note that all the vectors in $S$ can be represented as a conical combination of the column vectors of $A^*$. This is because $M$ is a non-negative matrix and the samples are $X = A^* M$. From (a), we know that all the column vectors of $A^*$ are also in $S$. Therefore, all the samples, except those that are scaled versions of $A^*$'s columns, can be written as a conical combination of the rest vectors in $S$.

We will prove (c) by contradiction. If a column vector of $A^*$ can be written as a conical combination of the rest vectors in $S$, then it means that this column vector can be represented as a conical combination of the column vectors in $A^*$. This will violate the fact that $A^*$ has full column rank. Hence, any column vector in $A^*$ cannot be represented as a conical sum of the rest vectors in $S$.   □

According to Claim 3, if $M$ satisfies the separability condition, and $A^*$ has full column rank, then Algorithm 4 can *exactly* recover $A^*$ (up to permutation and scaling of the column vectors in $A^*$). Once $A^*$ is recovered, estimating $W$ and $b$ is the same problem as learning one-layer ReLU generative model, which can be done by Algorithm 1. One problem with the above approach is that it requires the $M \in \mathbb{R}^{p \times n}$ matrix to satisfy the separability condition. This is true when, e.g., $W$ has full row rank, and the number of samples is $\Omega(2^k)$. Developing sample-efficient algorithms for more general generative models is definitely an interesting direction for future research.

We simulate Algorithm 4 on a two-layer generative model with $k = p = 5$ and $d = 10$. We generate $A^* \in \mathbb{R}^{10 \times 5}$ as a random Gaussian matrix, $W^* \in \mathbb{R}^{5 \times 5}$ as a random orthogonal matrix, and let $b^*$ be zero. Given $n$, we run 100 times of Algorithm 4, and each time we use a different set of random samples with size $n$. Table 1 lists the fraction of runs that Algorithm 4 successfully recovers $A^*$. We see that the probability of success increases as we are given more samples.

| Number of samples $n$ | 50 | 100 | 150 |
|---|---|---|---|
| Probability of success in 100 runs | 0.30 | 0.78 | 0.99 |

Table 1: We simulate a two-layer generative model: $A^* \in \mathbb{R}^{10 \times 5}$ is a random Gaussian matrix, $W^* \in \mathbb{R}^{5 \times 5}$ is a random orthogonal matrix, and $b^* = 0$. For a fixed number of samples, we run 100 times of Algorithm 4 with different input samples. This table shows the fraction of runs that Algorithm 4 successfully recovers $A^*$.

## H.3 Learning from Noisy Samples

It is an interesting direction to design algorithms that can learn from noisy samples, e.g., samples of the form $x = \mathrm{ReLU}(W^* z + b^*) + \xi$, where $\xi \sim \mathcal{N}(0, \sigma^2 I_d)$ represents the noise. In that case, Algorithm 1 would not work because both parts of our algorithm (i.e., learn from truncated samples, and estimate the pairwise angles) require clean samples. Nevertheless, the above problem is easy when $b^* = 0$. This is because we can estimate $\|W^*(i,:)\|_2$ using the fact that $\mathbb{E}_{z,\xi}[x(i)^2] = \|W^*(i,:)\|_2^2 / 2$, and estimate $\theta_{ij}^*$ using the following fact [CS09]:

$$\mathbb{E}_{z,\xi}[x(i)x(j)] = \frac{1}{2\pi} \|W^*(i,:)\|_2 \|W^*(j,:)\|_2 (\sin\left(\theta_{ij}^*\right) - (\pi - \theta_{ij}^*)\cos(\theta_{ij}^*)).$$

## Footnotes

[9]Here we define separability with respect to the $M$ matrix while [AGKM12, Definition 5.1] defines it with respect to the $A$ matrix, but they are equivalent definitions.