[Reviews · NeurIPS 2019]

Reviewer 1



A popular generative model these days is as follows: pass a standard Gaussian noise through a neural network. But a major unanswered question is what is the structure of the resulting distribution? Given samples from such a distribution, can we learn the distribution parameters? This question is the topic of this paper. Specifically, consider a 1-layer ReLU neural network, which is specified by a matrix W and a real bias b. Given a standard Gaussian as its input, the output of this networks is some complicated distribution. Given samples from this distribution, this paper gives an algorithm that learns the parameters W and b. Namely, they show if the number of samples is large enough, then they can recover W W^T and b approximately (Theorem 1), and they can also learn the target distribution within a small error in total variation distance (Corollary 1). The main assumption is that each component of b must be non-negative. Indeed, in Claim 2 a simple example is given demonstrating that if b has negative components, learning this negative component requires exponential sample complexity. This is the first paper that gives an algorithm with a rigorous guarantee for this important problem, and the paper is well-written. The technique of the paper is also interesting: they first learn the norms of the rows of the matrix W, and then they learn the angles between the rows. The main limitation of the paper is the assumption that each component of b is non-negative. It's really a limiting assumption, and there is no reason to believe that this will happen in practice. The authors have argued that if some component of b is negative, then to estimate this component we need exponential samples. But then maybe estimating the components of b is the wrong objective. In many applications, you just need to output some distribution that's close to the target in the total variation distance. Can we do this even if b has negative components? Other comments for the authors: - line 24: it's not clear what do you mean by "latent code z" - In equation (10) and elsewhere: write P(A) instead of E[1(A)] - Line 243: matrixl -> matrix - There are lower and upper bounds for the total variation distance between Gaussians, look at the arXiv preprint "The total variation distance between high-dimensional Gaussians" == After reading the response: thanks, I have increased my score.

Reviewer 2



This paper considers learning the distribution generated by one-layer ReLU functions, with input from Gaussian distribution. The high level idea is simple. First step is to estimate the bias term and the norm of each row of the weight vector. Second step is based on a well known fact on the covariance of two ReLU functions. The first step is shown via a maximum likelihood approach from an earlier work (DGTZ18). Overall, the paper is well presented and the proof looks correct to me. One limitation is that the proposed estimation method is somewhat restricted to the specific setting studied here. This makes the connection between this paper to the motivating examples (GANs, VAEs) somewhat weak. Other questions: *) Do similar results extend to more general input distributions? What are the boundaries beyond which estimation becomes hard? It may be worth discussing a little bit. *) In Sec 4.2: estimating the norm of W(i,:) and b(i) boils down to a single dimensional problem. I wonder if there is a simpler way to estimate these quantities.

Reviewer 3



The paper studies the problem of learning the parameters of a one-layer ReLU network by observing its outputs on a latent standard Gaussian input. This problem is different from the supervised learning problem typically studied for deep networks where we see both the input z to the network and the output y and wish to learn the parameters of the network theta. Here we know the input z is coming from a standard Gaussian and we only see the output y and we wish to learn the parameters of the network (which would translate to learning the rectified Gaussian distribution) The paper presents an exponential sample lower bound for learning if the bias is allowed to be arbitrary. This is because when the bias is allowed to be negative most of the realizations of the latent variable z can map to 0 in the output space due to rectification and hence we will see very few effective samples. Then the authors proceed to restrict the bias to be non-negative and using recent results on learning a Gaussian from truncated samples give an algorithm for learning the network when the weight matrix is well-conditioned. The main algorithmic tool used is projected gradient descent. The estimation proceeds in three steps. First we realize that we can have identifiable recovery only for WW^T. To recover WW^T, it suffices to have the two norms of each row of W and the angles between all pairs of rows. To recover the two norms of each row of W and the bias vector, the algorithm for learning from truncated samples is applied. Then to estimate the angle between rows, the paper uses a simple observation wherein the sign of inner products of two vectors u,v with a random Gaussian vector z is used to infer the angle between u,v. This estimation however carries the noise in the estimation from the previous step and one needs to show that the noise doesn’t accumulate by too much. Overall, the paper is well written and has clarity. I am not sure how novel the algorithm is given that is it heavily based on prior work but the idea of applying it to this setting carries merit. ========== POST AUTHOR-FEEDBACK: Thanks to the authors for their response where they explained the difficulties of extending the results to neural networks with larger number of layers. I will be staying with my current score.

[Author Response · NeurIPS 2019]

We thank all the reviewers for their time and valuable comments. For space limitation, we focus on addressing the main comments. **Reviewer #1** wants to see an algorithm that works when $b^*$ has negative values. We show below that our algorithm can still be used in that case to recover part of the parameters with small number of samples. Both **Reviewer #2** and **Reviewer #3** ask about generalization to other settings. We discuss below one possible approach to learn a two-layer generative model. Extending our results to more general settings is definitely an interesting direction and we hope that our current work can encourage more people to work on this important problem.

**Reviewer #1**

**"Provide an algorithm to output a distribution that's close to the target, even if $b$ has negative components."**

When $b^*$ has negative components, running our algorithm can still recover part of the parameters. Specifically, let $\Omega := \{i \in [d] : b^*(i) \geq 0\}$ be the set of coordinates that $b^*$ is non-negative, then the output of our algorithm $\widehat{b}$ and $\widehat{\Sigma}$ satisfies: 1) the sub-vector $\widehat{b}_\Omega$ is close to $b^*_\Omega$; 2) the sub-matrix $\widehat{\Sigma}_{\Omega \times \Omega}$ is close to $W^*_\Omega W^{*T}_\Omega$. This is because our algorithm only uses the $i$-th and $j$-th coordinates of the samples to estimate $\langle W^*(i,:), W^*(j,:) \rangle$ and $b^*(i), b^*(j)$. As a result, our guarantee (Theorem 1 in our paper) still holds for this part of the parameters. We will mention this in the paper.

For the rest part of the parameters, if the negative components of $b^*$ are small (in absolute value), then the error of our algorithm will be also small. Specifically, let $\Omega^c$ be the complement of $\Omega$. Suppose that $b^*(i) \geq -\eta \|W^*(i,:)\|_2$ for all $i \in \Omega^c$ and for some $\eta \geq 0$, then given $\widetilde{O}(\ln^2(d)/\epsilon^2)$ samples, the output of our algorithm satisfies $|\widehat{b}(i) - b^*(i)| \leq \max(\eta, \epsilon) \|W^*(i,:)\|_2$, for all $i \in \Omega^c$. One can show a similar result for $\langle W^*(i,:), W^*(j,:) \rangle$, for all $i \in \Omega^c$. We see that the error from negative bias is small if $\eta = O(\epsilon)$. If $\eta$ is large, i.e., if $b^*$ have large negative components, then estimating those parameters becomes difficult (as indicated by Claim 2 in our paper). In that case, maybe one should directly estimate the distribution (as suggested by the reviewer). This is an interesting direction for future research.

**Reviewer #2 and #3**

**"What happens when we increase the number of layers?"**

Besides the single-layer ReLU generative model considered in our paper, we also thought about extending our results to learning a two-layer generative model. Let $\mathcal{D}(A, W, b)$ be the distribution of a random variable $x \in \mathbb{R}^d$ defined by

$$x = A \operatorname{ReLU}(Wz + b), \text{ where } z \sim \mathcal{N}(0, I_k), A \in \mathbb{R}^{d \times p}, W \in \mathbb{R}^{p \times k}, b \in \mathbb{R}^p.$$

Given i.i.d. samples $x \sim \mathcal{D}(A, W, b)$, can we recover the parameters $A, W, b$ (up to permutation and scaling of the column vectors in $A$)? While this problem seems hard in general, we find an interesting connection between this problem and non-negative matrix factorization (NMF).

In MNF, we are given a non-negative matrix $X \in \mathbb{R}^{d \times n}$ and an integer $p > 0$, the goal is to find two non-negative matrices $A \in \mathbb{R}^{d \times p}, M \in \mathbb{R}^{p \times n}$ such that $X = AM$. This problem is NP-hard and [AGKM12] give the first polynomial-time algorithm under the "separability" assumption (Definition 5.1 in [AGKM12]).

In our problem, we are given $n$ samples $\{x_i\}_{i=1}^n$ from $\mathcal{D}(A, W, b)$. Stacking the samples gives a matrix $X \in \mathbb{R}^{d \times n}$:

$$X = AM, \text{ where } M(:,i) = \operatorname{ReLU}(Wz_i + b), i \in [n].$$

Note that $M \in \mathbb{R}^{p \times n}$ is non-negative while the entries of $A$ can have *arbitrary* sign. If $M$ satisfies the "separability" condition [AGKM12], and $A$ has full column rank (i.e., the columns of $A$ are linearly independent), then we can still use the same idea of [AGKM12] to *exactly* recover $A$ and $M$ (up to permutation and scaling of the column vectors in $A$). Once $M \in \mathbb{R}^{p \times n}$ is recovered, estimating $W$ and $b$ is the same problem as learning one-layer ReLU generative model, which can be done by our algorithm. One problem with the above approach is that it requires the $M \in \mathbb{R}^{p \times n}$ matrix to satisfy the "separability" condition. This is true when, e.g., $W$ has full row rank, and the number of samples is $\Omega(2^k)$. Developing sample-efficient algorithms for more general cases is definitely an important research direction.

**Reviewer #2**

**"Does similar results extend to more general input distributions?"**

This is an interesting research direction. In our paper we focus on the standard Gaussian distribution for two reasons: 1) It has already been used in VAEs, GANs, and reversible generative models as the input distribution; 2) Even for this simple input distribution, we already encountered some technical difficulties such as negative bias vector (see our response to Reviewer #1). It is not easy to directly extend our algorithm to other input distributions, but our high-level idea, i.e., first estimate the norm and then estimate the pairwise angle, may still be useful.

# References

[AGKM12] Arora, Sanjeev and Ge, Rong and Kannan, Ravindran and Moitra, Ankur. Computing a nonnegative matrix factorization–provably. *Forty-fourth Annual ACM Symposium on Theory of Computing (STOC)*, 2012.


[Meta-Review · NeurIPS 2019]

This is a cute paper giving a nice solution to a clean problem. It is somewhat stylized, but improves upon prior work; of course, the authors know that a more general problem and solution would be preferred... The reviewers provide a good deal of feedback that should be considered for final versions.